# Learning to Modulate pre-trained Models in RL

**Thomas Schmied**[1]**, Markus Hofmarcher**[2]**, Fabian Paischer**[1]**,**
**Razvan Pascanu**[3,4]**, Sepp Hochreiter**[1]
[1] ELLIS Unit Linz and LIT AI Lab, Institute for Machine Learning,
[2] JKU LIT SAL eSPML Lab, Institute for Machine Learning,
Johannes Kepler University, Linz, Austria
[3] Google DeepMind, [4] UCL
schmied@ml.jku.at

## Abstract

Reinforcement Learning (RL) has been successful in various domains like robotics, game playing, and simulation. While RL agents have shown impressive capabilities in their specific tasks, they insufficiently adapt to new tasks. In supervised learning, this adaptation problem is addressed by large-scale pre-training followed by fine-tuning to new down-stream tasks. Recently, pre-training on multiple tasks has been gaining traction in RL. However, fine-tuning a pre-trained model often suffers from catastrophic forgetting. That is, the performance on the pre-training tasks deteriorates when fine-tuning on new tasks. To investigate the catastrophic forgetting phenomenon, we first jointly pre-train a model on datasets from two benchmark suites, namely Meta-World and DMControl. Then, we evaluate and compare a variety of fine-tuning methods prevalent in natural language processing, both in terms of performance on new tasks, and how well performance on pre-training tasks is retained. Our study shows that with most fine-tuning approaches, the performance on pre-training tasks deteriorates significantly. Therefore, we propose a novel method, Learning-to-Modulate (L2M), that avoids the degradation of learned skills by modulating the information flow of the frozen pre-trained model via a learnable modulation pool. Our method achieves state-of-the-art performance on the Continual-World benchmark, while retaining performance on the pre-training tasks. Finally, to aid future research in this area, we release a dataset encompassing 50 Meta-World and 16 DMControl tasks.

## 1   Introduction

Reinforcement Learning (RL) has been instrumental in training agents capable of achieving notable successes, both in simulation, and in the real-world (Silver et al., 2016; Vinyals et al., 2019; Berner et al., 2019; Arjona-Medina et al., 2019; Bellemare et al., 2020; Degrave et al., 2022). However, such agents are usually highly specialized and incapable of performing well outside of a narrowly-defined task. Furthermore, adapting a pre-trained agent to a new task by fine-tuning usually results in decreased performance on prior tasks. This effect is well-known in the literature as *catastrophic forgetting* (McCloskey and Cohen, 1989).

A common paradigm to learn multiple tasks concurrently is multi-task learning (Caruana, 1997). However, typically, not all tasks we want an agent to learn are available at training time. In this case, new tasks must be learned in a sequential manner. Learning a new task ideally exploits knowledge from previously learned tasks and does not adversely affect the performance on these prior tasks. Recent works have demonstrated that models based on the Transformer architecture (Vaswani et al., 2017) excel at learning multiple tasks concurrently from large offline datasets (Lee et al., 2022; Reed et al., 2022; Jiang et al., 2022; Brohan et al., 2022; Gupta et al., 2022; Shridhar et al., 2022). We want

37th Conference on Neural Information Processing Systems (NeurIPS 2023).

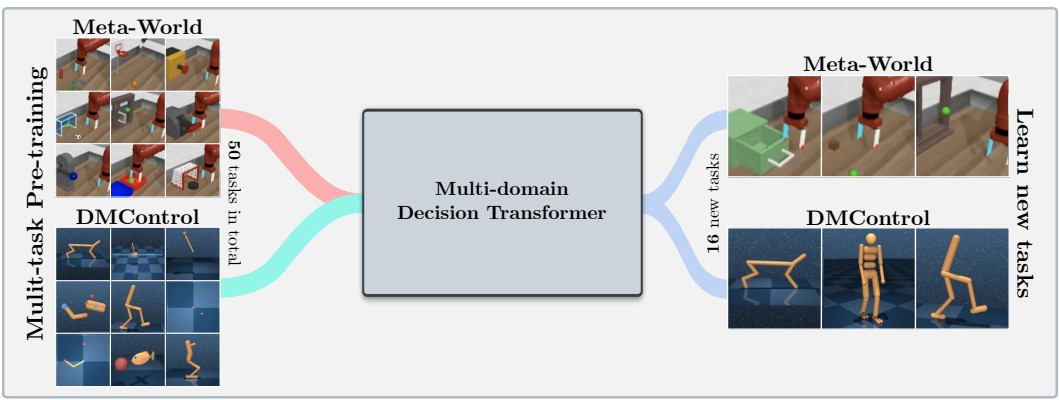

**Figure 1:** General setup of the experiments. We pre-train a Multi-Domain Decision Transformer (MDDT) model on a large set of tasks from multiple domains, namely Meta-World and DMControl. Then we fine-tune this model to learn additional tasks from these domains.

to utilize this capability in RL, thereby obtaining agents that can quickly learn new skills and adapt to new tasks. However, it is still unclear how an agent can learn new tasks and skills without negatively impacting its performance on the tasks it has already learned.

Our aim is to find methods that enable an agent to efficiently learn new tasks without compromising its proficiency in previously acquired tasks. In this regard, we draw inspiration from fine-tuning (FT) techniques prevalent in supervised learning, such as parameter-efficient fine-tuning (PEFT, (Houlsby et al., 2019; Hu et al., 2022; Liu et al., 2022b)) and prompt-based tuning (PBT, (Lester et al., 2021; Li and Liang, 2021)). Both PBT, and PEFT, incorporate a small set of new parameters to adapt a pre-trained model to new tasks at low cost. Thus, they intrinsically avoid catastrophic forgetting. However, it is unclear how well these methods can be adopted for training RL agents from offline datasets. Therefore, we first conduct a comprehensive evaluation for fine-tuning a pretrained Decision Transformer (DT, Chen et al., 2021) on two established RL benchmarks, namely Meta-World (Yu et al., 2020b; Wolczyk et al., 2021) and DMControl (Tassa et al., 2018). We first pre-train a DT jointly on tasks from both domains, then we transfer the pre-trained model to new tasks using various FT, PEFT and PBT methods (Figure 1). We find that FT methods adjust well to new tasks, but performance on previous tasks generally decreases. PBT, in contrast, retains performance on previous tasks but does not adapt well to new tasks.

To combine the advantages of both approaches, we propose Learning-to-Modulate (L2M), which is based on two recent works from NLP and computer vision, namely Learning-to-prompt (L2P, Wang et al., 2022c) and Low Rank Adaptation (LoRA, Hu et al., 2022). L2M operates in a task-agnostic manner and efficiently adapts a pre-trained model to new tasks via a learnable modulation pool. The modulation pool consists of learnable keys that are associated with learnable modulation matrices. Based on the current context, L2M selects a set of modulators which are learned during fine-tuning on the new task. This assumes that tasks can be well discriminated, such that suitable modulators are selected. Indeed, we observe an emergent clustering of tasks in the embedding layer of our model after pre-training, as illustrated in Figure 2. In turn, L2M is capable of quickly adapting to new tasks, while retaining performance on prior tasks, and introduces only minimal additional parameters per task relative to the full model size.

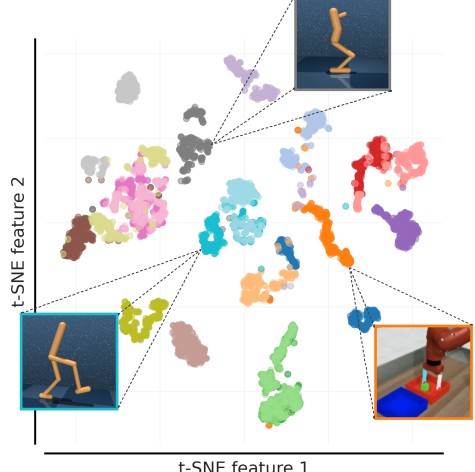

**Figure 2:** t-SNE clustering of state embeddings for the first ten MT40 and DMControl tasks. Similar tasks are clustered together, while dissimilar tasks are apart (legend is available in Appendix F).

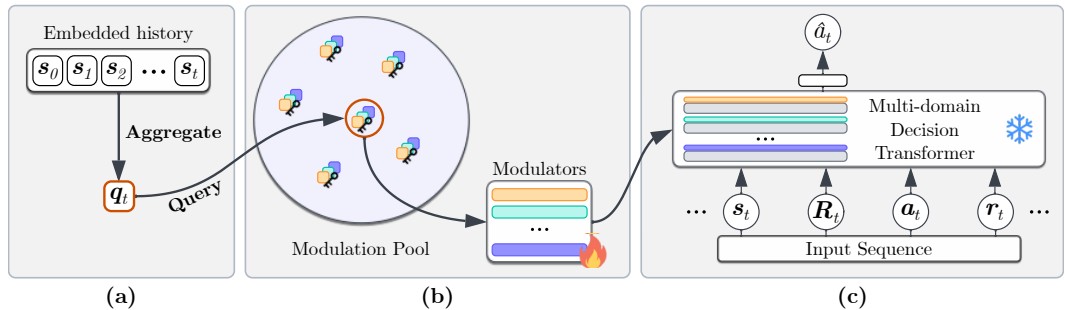

**Figure 3:** Illustration of L2M. **(a)** We construct a query $\mathbf{q}_t$ by aggregating the embedded history of state-tokens. **(b)** The query $\mathbf{q}_t$ is matched with learnable keys in a modulation pool, which map to learnable modulation matrices. We retrieve the modulation matrices with the highest similarity between $\mathbf{q}_t$ and every key in the modulation pool. **(c)** Retrieved modulation matrices modify the pre-trained and frozen multi-domain DT.

To summarize, we make the following **contributions**:

- We conduct an extensive evaluation of fine-tuning, parameter-efficient fine-tuning, and prompting methods for Transformers in RL.
- We propose the novel L2M for efficient fine-tuning of a frozen pre-trained model by modulating the information flow via learnable modulation matrices.
- We release a dataset of trajectories for the Meta-World and DMControl benchmark suites.

## 2 Method

We propose L2M, a method to adapt a pre-trained RL agent such that new tasks can be learned efficiently without compromising performance on pre-training tasks. To this end, we combine approaches from prompt-based tuning and parameter-efficient fine-tuning.

### 2.1 Background

**Reinforcement Learning.** We consider a Markov decision process (MDP) represented by the tuple $(\mathcal{S}, \mathcal{A}, \mathcal{P}, \mathcal{R})$. $\mathcal{S}$ and $\mathcal{A}$ denote state and action spaces, respectively, with states $s_t \in \mathcal{S}$ and actions $a_t \in \mathcal{A}$ at timestep $t$. The transition function $\mathcal{P}(s_{t+1} \mid s_t, a_t)$ takes a state-action pair and yields a probability distribution over next states. The reward function $\mathcal{R}(r_t \mid s_t, a_t)$ maps a given state-action pair to a scalar reward $r_t$. The objective is to learn a policy $\pi(a_t \mid s_t)$ that predicts an action $a_t$ given $s_t$ that maximizes the reward $r_t$.

**Decision Transformer.** Decision Transformer (DT, Chen et al., 2021) rephrases RL as a sequence modelling problem. This assumes access to a dataset $\mathcal{D} = \{\tau_i \mid 1 \leq i \leq N\}$, where $\tau_i = (s_0, a_0, r_0, \ldots, s_T, a_T, r_T)$ represents a trajectory of length $T$ consisting of state-action-reward triplets and $N$ defines the number of trajectories in $\mathcal{D}$. Further, we augment each trajectory $\tau_i$ with *returns-to-go* $\hat{R}_t = \sum_{t'=t}^{T} r_{t'}$. The policy $\pi_\theta(a_t \mid s_{t-C:t}, \hat{R}_{t-C:t}, a_{t-C:t-1}, r_{t-C:t-1})$, with parameters $\theta$ and context length $C$, is then trained in a supervised manner via upside-down-RL (Schmidhuber, 2019), minimising the cross-entropy $\mathcal{L}_{CE}(\hat{a}_t, a_t)$ between predicted action $\hat{a}_t$ and ground-truth action $a_t$. During inference, the policy produces a sequence of actions conditioned on a high return-to-go.

**Low Rank Adaptation (LoRA).** LoRA (Hu et al., 2022) is a parameter efficient method for fine-tuning pre-trained models. It performs a low-rank decomposition of a weight matrix that is utilized to modulate the information flow in the pre-trained model. Particularly, we assume a pretrained layer $l$ with weight matrix $\boldsymbol{W}_l \in \mathbb{R}^{d_{\text{in}} \times d_{\text{out}}}$ that receives $\boldsymbol{x}_{l-1} \in \mathbb{R}^{1 \times d_{\text{in}}}$ as input. LoRA adds learnable low rank modulation matrices $\boldsymbol{A}_l \in \mathbb{R}^{d_{\text{in}} \times r}$ and $\boldsymbol{B}_l \in \mathbb{R}^{r \times d_{\text{out}}}$, where $r \ll d_{\text{in}}$. The output $\boldsymbol{h}_l$ of layer $l$ is then computed as:

$$\boldsymbol{h}_l = \boldsymbol{W}_l \boldsymbol{x}_{l-1} + \boldsymbol{B}_l \boldsymbol{A}_l \boldsymbol{x}_{l-1}. \tag{1}$$

During the fine-tuning stage, only the parameters of $\boldsymbol{A}_l$ and $\boldsymbol{B}_l$ are updated.

## 2.2 Learning-to-Modulate (L2M)

We propose L2M, a method that combines the advantages of PEFT and PBT approaches. In particular, we rely on LoRA (Hu et al., 2022), and refer to the trainable parameters it induces as modulators of the pre-trained model. Further, we draw inspiration from Learning-to-prompt (L2P, Wang et al., 2022c), and maintain a pool of modulators associated with learnable keys. For a given input sequence, we retrieve the best modulator weights from the pool, and leverage them to alter the behaviour of the pretrained model (Figure 3).

We define a *modulation pool* that contains a set of $M$ learnable keys, $\mathbf{K}_{\text{pool}} = \{\mathbf{k}_i \mid 1 \leq i \leq M\}$. All keys in $\mathbf{K}_{\text{pool}}$ are initialized uniformly from $[-1, 1]$. Each $\mathbf{k}_i$ is associated with a set of modulation matrices $\{\mathbf{A}_b^i, \mathbf{B}_b^i \mid 1 \leq b \leq B\}$, for each layer block $b$ of a DT with $B$ layer blocks. As in Hu et al. (2022), we initialize $\mathbf{A}$ according to a normal distribution around zero and $\mathbf{B}$ with zeros. Thus, no modulation can occur at the beginning of training.

At time $t$, we generate a query vector $\mathbf{q}_t \in \mathbb{R}^d$, where $d$ is the embedding dimension of the pre-trained model, in the following way. First, we construct a matrix $\mathbf{S}_{t-C:t} \in \mathbb{R}^{C \times d}$ from the states of a trajectory $\tau$ with the context length $C$, after they are processed via the embedding layer of the frozen pre-trained model. Then, we reduce the matrix to a query vector $\mathbf{q}_t$ by an aggregation function $g(\cdot)$:

$$\mathbf{q}_t = g(\mathbf{S}_{t-C:t}) \tag{2}$$

where we use mean-pooling for $g(\cdot)$. We retrieve a set of modulation matrices $\{\mathbf{A}_b^j, \mathbf{B}_b^j \mid 1 \leq b \leq B\}$ by the maximum similarity between each $\mathbf{k}_i \in \mathbf{K}_{\text{pool}}$ and the query $\mathbf{q}_t$ at timestep $t$:

$$j = \underset{\mathbf{k}_i \in \mathbf{K}_{\text{pool}}}{\arg\max} \ \text{sim}(\mathbf{q}_t, \mathbf{k}_i) \, n(\mathbf{k}_i)^{-1} \tag{3}$$

In our case, $\text{sim}(\cdot, \cdot)$ corresponds to the cosine similarity and $n(\mathbf{k}_i)$ represents the selection count for key $\mathbf{k}_i$. Every time a key $\mathbf{k}_i$ is selected, we increase $n(\mathbf{k}_i)$ by one. This way, we discourage that different queries always attend to the same key.

We use $\{\mathbf{A}_b^j, \mathbf{B}_b^j \mid 1 \leq b \leq B\}$ to modulate the pre-trained model according to Equation (1). More specifically, we apply LoRA on the queries and values in the self-attention mechanism, as well as on the activation in the feed-forward block of the pre-trained model. Only the modulators are learned via gradient descent, while the pre-trained model remains frozen. Following Wang et al. (2022c), the key is updated by maximizing the cosine similarity between $\mathbf{q}_t$ and $\mathbf{k}_j$ via an additional term in the end-to-end objective:

$$\min_{\theta, \mathbf{k}_i} \mathcal{L}_{CE}(\hat{a}_t, a_t) - \lambda \ \text{sim}(\text{stopgrad}(\mathbf{q}_t), \mathbf{k}_j) \tag{4}$$

where $\lambda$ is a hyperparameter.

L2M unifies the benefits of LoRA and L2P, ensuring high-performance and few additional learnable parameters, while also avoiding forgetting on the pre-trained tasks.

## 2.3 Multi-Domain Decision Transformer (MDDT)

We extend the Decision Transformer architecture proposed by Chen et al. (2021) to handle inputs from multiple domains with varying state/action spaces. Since the dimensionality of states differ between Meta-World and DMControl we construct a unified state space, that comprises all dimension of DMControl and Meta-World environments. This state-space consists of 204 dimensions in total. Dimensions that are unused by a certain task are padded with zeros. Finally, the states are embedded with a linear layer before serving as input to the MDDT. Similar to Reed et al. (2022) and Brohan et al. (2022), we tokenise each action dimension and autoregressively predict action tokens. In the environments we consider, all actions are bounded by [-1, 1]. We use a min-max tokenisation, which comprises min-max normalisation and subsequent uniform discretisation into 64 bins. These changes enable processing observations and actions originating from different environments at training and test time. In line with prior work (Chen et al., 2021; Lee et al., 2022), we train the DT via return-conditioned upside-down RL using a cross-entropy loss to predict next actions autoregressively. We set the target return to the maximum observed return in the respective dataset, but also found that constant proxies per domain work well. We provide further implementation details on our MDDT as well as a discussion on limitations and implications of our design choices in Appendix C.

# 3 Experiments

We consider two different benchmark suites that comprise 66 different tasks in total, namely Meta-World (Yu et al., 2020b; Wolczyk et al., 2021) and DMControl (Tassa et al., 2018). Meta-World consists of 50 diverse tasks for robotic manipulation, such as grasping, manipulating objects, opening/closing a window, pushing buttons, locking/unlocking a door, and throwing a basketball. We follow Wolczyk et al. (2021) and split them into 40 pre-training (MT40), and 10 fine-tuning tasks (CW10). We use Meta-World v2, instead of v1 used by Wolczyk et al. (2021), but will refer to it as Meta-World throughout this work. Further, we select 16 tasks from DMControl and assign 10 tasks to the pre-training set (DMC10) and 6 tasks to the fine-tuning set (DMC6). We elaborate on the different environments and on the data collection procedure in Appendix A, and Appendix D, respectively. Unlike Meta-World, which shares the same state and action spaces across tasks ($|\mathcal{S}| = 39, |\mathcal{A}| = 6$), the state and action spaces of DMControl vary for each task (e.g., for hopper $|\mathcal{S}| = 15, |\mathcal{A}| = 4$, for cheetah $|\mathcal{S}| = 17, |\mathcal{A}| = 6$), as listed in Appendix D.

We collect a dataset[1] by training task-specific agents on each task using an agent based on Soft Actor Critic (SAC) (Haarnoja et al., 2018). The average performance for SAC across the Meta-World and DMControl task splits is shown in Table 1. For each of the 50 tasks in Meta-World, we include 10K trajectories of length 200 in the dataset, amounting to 100M transitions in total. For DMControl, we include 1000 trajectories of length 1000 for each task, amounting to 16M transitions in total. The datasets contain the entire replay buffer of the task-specific agents, and consequently behaviours ranging from random to expert. We choose this collection scheme since prior work has illustrated the benefits of training agents on data that comprises mixed behaviour (Lee et al., 2022). We give further details on the training procedure, including hyperparameters and learning curves in Appendix D.

**Table 1:** Performance measures of task specific SAC-based agents used for data collection averaged over the different task splits. Mean and standard deviation are shown.

<table>
<tr><td colspan="3" align="center">(a) Meta-World</td><td colspan="2" align="center">(b) DMControl</td></tr>
<tr><td>Dataset</td><td>Success Rate</td><td>Mean Reward</td><td>Dataset</td><td>Mean Reward</td></tr>
<tr><td>MT40</td><td>0.84 ± 0.34</td><td>1414.62 ± 439.39</td><td>DMC10</td><td>788.36 ± 219.11</td></tr>
<tr><td>CW10</td><td>1.0 ± 0.0</td><td>1540.49 ± 184.43</td><td>DMC6</td><td>840.37 ± 216.63</td></tr>
</table>

Figure 1 illustrates our experimental design. First, we pre-train a MDDT on all pre-training datasets (MT40 and DMC10) simultaneously (see Section 3.1). Next, we conduct a broad evaluation of different FT, PEFT and PBT methods to adapt the pre-trained model to each of the held-out fine-tuning tasks (CW10 and DMC6). In Section 3.2, we show the results for fine-tuning on all CW10 and DMC6 tasks individually. Finally, we present results for training all methods in a continual RL setup, in which tasks are introduced sequentially (Section 3.3). All our models are trained offline on the generated datasets and evaluated online in the respective environments.

Unless mentioned otherwise, we report IQM and 95% bootstrapped confidence intervals, as proposed by Agarwal et al. (2021) (3 seeds per method and 50K bootstrap samples). We report success rates and data-normalized scores (normalized by mean score in dataset and random agent performance as suggested by Fu et al. (2020)) for Meta-World and DMControl, respectively.

## 3.1 Pre-training

We pre-train a MDDT on our datasets collected for MT40 and DMC10. This already results in strong performance on the pre-training tasks (60% for MT40, 94% for DMC10). We experiment with different model sizes (40M to 200M parameters), varying number of Transformer layers, number of heads per layer and embedding dimension. Generally, there is a trend that more complex models perform better. For all our subsequent experiments, we use a 40M parameter model, as it achieves a good trade-off between performance and training time. Performance scores, learning curves, implementation details and hyperparameters for our pre-training experiments are provided in Appendix E.

---

[1]Source code and datasets are available at: `https://github.com/ml-jku/L2M`

After pre-training, we analyse the learned representations of the model via t-SNE (Van der Maaten and Hinton, 2008). Figure 2 illustrates emerging clusters of learned state embeddings for ten tasks of MT40 and DMC10. We observe that similar tasks cluster together (e.g., *button-press-v2* and *button-press-wall-v2*), whereas dissimilar tasks are well separated (e.g., *assembly-v2* and *bin-picking-v2*). We repeat this analysis for rewards, RTG and action token embeddings and explain the clustering procedure in Appendix F.

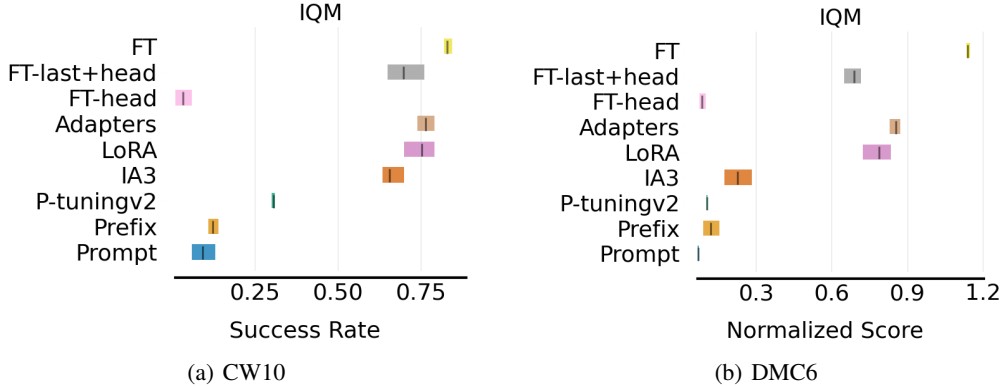

(a) CW10        (b) DMC6

**Figure 4:** IQM and 95% CIs for single-task fine-tuning on **(a)** CW10 and **(b)** DMC6. Models are pre-trained on MT40/DMC6 and are then optimized for each CW10/DMC6 task individually.

### 3.2 Single-Task Fine-Tuning

We evaluate the performance of various FT, PEFT and PBT strategies for fine-tuning on the held-out tasks. Overall, we compare a total of 9 methods: Full fine-tuning (FT), FT of action head (FT-head), FT of last Transformer layer and action head (FT-last+head), Adapters (Houlsby et al., 2019), LoRA (Hu et al., 2022), (IA)³ (Liu et al., 2022b), Prompt-tuning (Lester et al., 2021), Prefix-tuning (Li and Liang, 2021), and P-tuning v2 (Liu et al., 2021b). We do not compare against meta-RL algorithms, as Mandi et al. (2022) showed that FT approaches perform on-par or better on several tasks. A detailed list of hyperparameters, training details and description for each method are provided in Appendix G. We fine-tune the pre-trained model for each task individually and aggregate the performance over all tasks within each domain.

In Figure 4, we show IQM over all tasks for all methods on CW10 and DMC6. Overall, FT attains the highest scores, since it can utilize the entire model capacity. Thus, it serves as an upper bound in terms of performance. Adapters achieve the second-highest scores on average, followed by LoRA, (IA)³ and FT-last+head. Notably, there is a large gap between PBT and PEFT methods, particularly on CW10. Fine-tuning the action head for each task separately, results in the worst performance. While the absolute scores differ between the two domains, the relative performance ranking among methods remains equivalent. On DMC6, we observe a larger performance gap between full FT and PEFT methods. While in Meta-World only the locomotion differs between tasks, in DMControl the entire robot morphology may change.

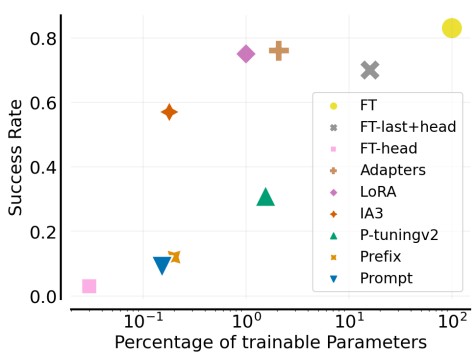

**Figure 5:** Success rate vs. fraction of parameters trained for various fine-tuning techniques on single-task experiments for CW10.

Therefore, more capacity of the model is required to sufficiently adapt to the new tasks.

We highlight the efficacy of the considered methods by comparing the fraction of trainable parameters against attained performance on CW10 in Figure 5 (see Appendix G for same comparison on DMC6). FT updates all parameters, while Adapters update two orders of magnitude less parameters (2.1%).

In contrast, LoRA requires less than half the amount of parameters of Adapters (1%), while reaching a similar level of performance. Notably, (IA)[3], trains approximately the same amount of parameters as Prompt-tuning (0.21%), but attains significantly higher performance. This result indicates that we can achieve decent performance with a small amount of trainable parameters if the pre-training and fine-tuning tasks are similar. Overall, LoRA compares favourably in both dimensions.

## 3.3 Continual Fine-Tuning

Ultimately, our goal is to adapt the pre-trained model to multiple novel tasks, while alleviating forgetting of tasks learned during pre-training. In this regard, we adapt the pre-trained model to all fine-tuning tasks in a sequential manner, as proposed by Wolczyk et al. (2021) for CW10. Moreover, we evaluate forgetting by measuring the performance on tasks acquired during pre-training after fine-tuning.

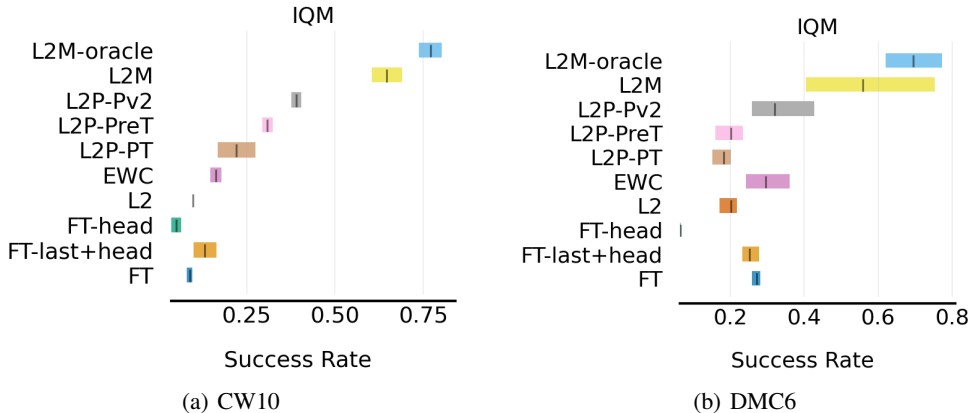

(a) CW10  (b) DMC6

**Figure 6:** IQM and 95% CIs of success rates for CRL experiments on **(a)** CW10 and **(b)** DMC6. Models are pre-trained on MT40+DMC10 and are then trained on the tasks from CW10/DMC6 sequentially. On each task, we train for 100K steps and then move to the next task in the sequence.

Again, we compare the following fine-tuning variations: FT, FT-head, and FT-last+head. Additionally, we augment the PBT methods of the previous section with a prompt pool as in L2P, which enables learning of task-specific prompts. We refer to these methods as L2P + Prompt-tuning (L2P-PT), L2P + Prefix-tuning (L2P-PreT), L2P + P-tuning v2 (L2P-Pv2). We note that L2P-PT reflects the original version of L2P proposed by Wang et al. (2022c) which was originally applied in computer vision. Additionally, we compare L2M to two established methods from Continual RL (CRL), namely Elastic Weight Consolidation (EWC), and L2, both propposed by Kirkpatrick et al. (2017). EWC constrains task-specific weights to not change too much when adapting to a new task by estimating their individual importances. L2 applies a uniform constraint to all weights without considering their relevance. Moreover, we add another implementation of L2M, which is equipped with an oracle that provides information on what task is currently being observed in terms of a task index (L2M-Oracle). For L2M-oracle, the modulation pool contains as many modulators as there are tasks. At training time, the task index refers to the dataset the batches are sampled from. At inference time, the task index refers to the environment the DT is currently evaluated in. In contrast to Wolczyk et al. (2021), we do not use separate action heads per task for L2M to remain task agnostic. To illustrate an upper bound on performance, we provide scores for two multi-task baselines, which train on all tasks simultaneously, either from scratch, or after pre-training (Table 7 in Appendix H). We train each method for 100K steps per task and evaluate on all tasks every 50K steps. We provide further training details and hyperparameters in Appendix H.

In Figures 6(a) and 6(b), we show the performance scores of all methods on CW10 and DMC6, respectively. In addition, we report forgetting scores (as calculated by Wolczyk et al. (2021)), and rewards obtained at the end of training, in Appendix H. L2M outperforms all other approaches in both domains, attaining an average success rate of 65% across CW10 tasks and a normalized score of 56% across DMC6 tasks. Adding a task oracle to L2M increases the success rate to 76% and 70% on CW10 and DMC6, respectively, and closely matches the single-task performance of LoRA. L2P

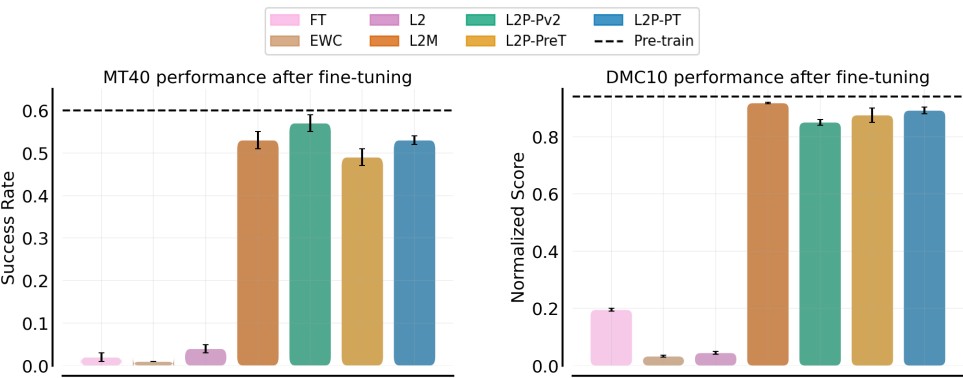

**Figure 7:** Performance on the MT40 and DMC10 pre-training tasks after fine-tuning.

combined with different prompting approaches performs considerably worse than L2M. Interestingly, the established CRL method EWC does not mitigate forgetting sufficiently. Similar results for EWC were reported by Ben-Iwhiwhu et al. (2022) in an online RL setting. To the best of our knowledge, the results of our method, L2M, are the highest reported results on Continual-World v2 to date. Prior work reports average success rates of roughly 40% (Caccia et al., 2022; Ben-Iwhiwhu et al., 2022). Even though L2M outperforms other approaches, there is a considerable gap between L2M and L2M-oracle. We surmise this is due to conflation effects in the prompt pool and aim to investigate this in more detail in future work.

**Performance on pre-training tasks.** Finally, we evaluate performance on the pre-training tasks after fine-tuning, as shown in Figure 7. To remain task-agnostic, we train a set of 100 keys for the pre-training tasks, which we concatenate to the set of keys introduced during the fine-tuning stage. While FT, L2 and EWC experience a severe drop in performance, L2M and L2P-based approaches maintain a similar performance level as prior to fine-tuning. Thus, L2M preserves performance on the pre-training tasks, while it effectively adapts to new tasks.

## 3.4 Ablation Studies

To gain a better understanding of our method and its limitations, we conduct a number of additional ablations.

**Modulation Targets.** One important design choice in L2M is which weights of the pre-trained model to modulate. In principle, LoRA can be applied to any weight matrix in the pre-trained model. By default, we employ LoRA on the queries and values in the self-attention mechanism, and on the position-wise feedforward layer. In Appendix G.1, we conduct an ablation study in which we vary the modulation targets. Our analysis suggests that modulating the attention mechanism is not as important as modulating the position-wise feed-forward layer. However, modulating both, the attention mechanism, and the position-wise feed-forward layer yields the best results. Nevertheless, this performance gain comes at the cost of more trainable parameters.

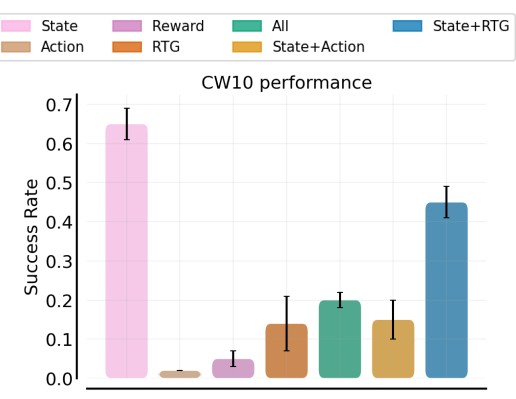

**Figure 8:** Aggregation token ablation on CW10.

**Query Representation.** As specified in Section 2.2, we use embedded state tokens aggregated over a specified context window as input query to the prompt pool in L2M. This design choice is inspired by the observed task separation in the embedding layer after pre-training (see Figure 2). We investigate other choices of individual tokens and token combinations (state, action, reward, RTG, state-action, state-RTG) to represent the query (see Figure 8). Indeed, we observe that using the state embeddings

results in the best overall performance, as it aids task separation. In contrast, using RTG or action embeddings deteriorates performance. Furthermore, we also investigate the effect of using different layer representations and context windows to construct the query in Appendix H.1.

**Alternative Modulators.** We conduct an ablation study in which we compare L2M against L2M in combination with $(IA)^3$ in Appendix H.2. Instead of the low-rank modulation matrices, $(IA)^3$ employs elementwise multiplication with learnable modulation vectors. While performance decreases with L2M-$(IA)^3$, it compares favourably in terms of parameter-efficiency. Depending on the task, this may be preferred.

**Single-domain DT on Meta-World.** To better understand the potential drawbacks of using a multi-domain model, we pre-train and fine-tune a single-domain model only on Meta-World (see Appendix H.3). Due to the common state and action spaces in Meta-World, we use a simpler, non-discretized, action space and training objective (MSE) for this experiment. This specialised single-domain model obtains considerably higher performance scores. However, these performance gains come at the cost of the loss of generality, as the specialised model can only handle the particular state/action space it was trained on. Thus, fine-tuning it to tasks with new state/action spaces is not possible. Nevertheless, this experiment highlights the drawbacks of our current mechanisms, such as action discretisation and autoregressive action-prediction, that enable multi-domain training.

**Cross-domain FT.** Finally, to better understand the potential upsides of using a multi-domain model, we pre-train a DT (with unified state space and action discretisation) on Meta-World only (MT40) and then fine-tune it on DMControl (DMC6). We observe that the fine-tuning performance on DMC6 (different domain) is considerably worse than for the MDDT (see Appendix H.4). In addition, we also fine-tune the pre-trained single-domain model on CW10 (same domain). Interestingly, the final performance on CW10 is also lower compared to the MDDT that was pre-trained on both domains. This experiment indicates, that multi-domain pre-training can, indeed, have a positive effect on the fine-tuning performance.

We provide additional details on our ablation studies in Appendix G and H.

## 4   Related Work

**Transformers in RL.** Since the inception of Transformers (Vaswani et al., 2017) there has been a widespread adoption of the underlying architecture in various areas such as NLP (Devlin et al., 2019; Radford et al., 2019; Brown et al., 2020), computer vision (Dosovitskiy et al., 2021; He et al., 2022; Radford et al., 2021; Fürst et al., 2022; Ramesh et al., 2021; Rombach et al., 2022), speech recognition (Radford et al., 2022; Baevski et al., 2020) or video generation (Ho et al., 2022; Singer et al., 2022). More recently, Transformers have found their way into RL. Similar to the DT Chen et al. (2021), Trajectory Transformer (Janner et al., 2021) is based on the GPT architecture (Radford et al., 2018), but relies on dynamics modelling. Lee et al. (2022) extended DT to a multi-game setup to learn to play 46 Atari games. Meanwhile, a variety of DT-variants have been proposed (Zheng et al., 2022; Wang et al., 2022a; Shang et al., 2022; Meng et al., 2021). Siebenborn et al. (2022) replace the Transformer in DT with an LSTM (Hochreiter and Schmidhuber, 1997). PromptDT (Xu et al., 2022b) demonstrated that prompting a pre-trained DT model with expert trajectories can improve the agent's ability to generalize to new tasks. Jiang et al. (2022) presented a prompt-based Transformer for robot manipulation, that integrates multi-modal prompts via cross-attention. Xu et al. (2022a) propose to augment a DT with a hyper-network for parameter-efficient policy adaptation. A number of other works (which are largely orthogonal to our approach) instead aim to improve the pre-training stage, for example by predicting masked-out parts of the sequence (Carroll et al., 2022; Liu et al., 2022a; Sun et al., 2022). Reed et al. (2022) trained a Transformer that scaled to over 600 tasks. Most recently, Brohan et al. (2022) presented a scalable Transformer for real-world robotics manipulation. Laskin et al. (2022) and Liu and Abbeel (2023) make use of a multi-episodic context to elicit in-context learning capabilities. Other works use a Transformer backbone for history compression in online RL (Parisotto et al., 2020; Paischer et al., 2022, 2023). Li et al. (2023) and Yang et al. (2023) cover the landscape of Transformers in RL in more detail.

**Continual and multi-task RL.** A plethora of works have considered learning multiple tasks concurrently via RL (Tanaka and Yamamura, 2003; Rusu et al., 2016a; Borsa et al., 2016; Rajeswaran et al., 2017; El Bsat et al., 2017; Andreas et al., 2017; D'Eramo et al., 2020; Yu et al., 2020a; Igl et al., 2020; Sodhani et al., 2021). In CRL, however, tasks are considered to not be readily available at the same

time. Many continual learning methods were proposed for computer vision, but can also be applied to a CRL setting. Early works include regularization approaches, such as EWC (Kirkpatrick et al., 2017), Synaptic Intelligence (Zenke et al., 2017), Memory-aware Synapses (Aljundi et al., 2018), Gradient Episodic Memory (GEM, Lopez-Paz and Ranzato, 2017) and A-GEM (Chaudhry et al., 2019). Other approaches rely on adding new modules to existing architecture (Rusu et al., 2016b), iterative pruning (Mallya and Lazebnik, 2018), or improved exploration (Steinparz et al., 2022). A number of approaches have been tested on the Continual-World benchmark (Wolczyk et al., 2021), including 3RL (Caccia et al., 2022), ClonExSAC (Wolczyk et al., 2022) and Modulating masks (Ben-Iwhiwhu et al., 2022). L2P (Wang et al., 2022c) learns to prompt a frozen Vision Transformer (Dosovitskiy et al., 2021) and consistently outperforms prior methods on a number of continual learning benchmarks. Other follow-up works that rely on a prompting mechanism for CL have been proposed (Wang et al., 2022b; Smith et al., 2022; Razdaibiedina et al., 2023). Recent works provide a comprehensive overview of the field of CRL (Hadsell et al., 2020; Lesort et al., 2020; Khetarpal et al., 2022; Baker et al., 2023).

**Parameter-efficient fine-tuning and Prompting.** Large-language models (LLMs) are usually pre-trained on vast amounts of data (Devlin et al., 2019; Radford et al., 2019; Brown et al., 2020). After pre-training, it is desirable to specialize or fine-tune the foundation model (Bommasani et al., 2021) to a down-stream task. A common way is fine-tuning all network weights or a fraction thereof. Fine-tuning the entire network is costly and suffers from catastrophic forgetting. Parameter-efficient fine-tuning methods and prompt-based tuning offer attractive alternatives. Houlsby et al. (2019) repurposed Adapter modules (Rebuffi et al., 2017) to interleave pretrained Transformer layers. Variations thereof have been proposed (Bapna et al., 2019; Pfeiffer et al., 2021, 2020; Karimi Mahabadi et al., 2021). Low Rank Adaptation injects trainable low-rank decomposition matrices into every layer of the model (Hu et al., 2022). $(IA)^3$ modulates the inner activation flow of the Transformer layers by elementwise multiplication with learned modulation vectors (Liu et al., 2022b). Prompt tuning conditions the pre-trained model on new tasks by prepending learnable prompts to the embedded input sequence (Lester et al., 2021). Similarly, prefix-tuning adds learnable prefix vectors to the keys and values of each attention head input (Li and Liang, 2021). P-tuning v2, applies learnable prompts at each Transformer layer, instead of merely the input layer (Liu et al., 2021b). UniPELT combines different PEFT methods in a unified framework (Mao et al., 2022) Liu et al. (2021a) give a comprehensive overview of prompt-based learning for NLP.

# 5   Conclusion

Adapting agents to new tasks, while preserving performance on previously learned tasks, remains a major challenge towards more general RL agents. Consequently, we conduct a comprehensive evaluation of established fine-tuning, PEFT and PBT methods for Transformers in RL. We evaluate both how well new tasks are learned and how strongly performance deteriorates on the pre-training tasks. While full fine-tuning of a pre-trained model adapts well to new tasks, it suffers from catastrophic forgetting of pre-training tasks. Prompt-based tuning techniques preserve performance on pre-training tasks, but cannot compete on new tasks. We propose a novel method, L2M, which performs well in both dimensions. L2M enables efficient fine-tuning of pre-trained models via learnable modulation matrices, resulting in high performance both on new tasks and on pre-training tasks. Finally, we release a large dataset of trajectories for the Meta-World and DMControl benchmark suites to facilitate future research in the domain of offline RL.

For future work, we envision an investigation on the compositionality of learned modulation matrices. While currently a distinct set of modulators is selected for a given input sequence, we believe that by combining modulators across tasks, our method can improve in terms of performance and parameter-efficiency. Such an approach could improve forward-transfer, where new tasks benefit even more from previously learned tasks by combining existing skills. Furthermore, the context length of the models we consider is currently limited. Extending the context to leverage information from multiple episodes and tasks may improve the model's ability to generalize to new tasks or domains. Another promising avenue of research is to investigate more diverse domains and explore how different fine-tuning methods perform on more distinct fine-tuning domains.

## Acknowledgments and Disclosure of Funding

The ELLIS Unit Linz, the LIT AI Lab, the Institute for Machine Learning, are supported by the Federal State Upper Austria. We thank the projects AI-MOTION (LIT-2018-6-YOU-212), DeepFlood (LIT-2019-8-YOU-213), Medical Cognitive Computing Center (MC3), INCONTROL-RL (FFG-881064), PRIMAL (FFG-873979), S3AI (FFG-872172), DL for GranularFlow (FFG-871302), EPILEPSIA (FFG-892171), AIRI FG 9-N (FWF-36284, FWF-36235), AI4GreenHeatingGrids(FFG- 899943), INTEGRATE (FFG-892418), ELISE (H2020-ICT-2019-3 ID: 951847), Stars4Waters (HORIZON-CL6-2021-CLIMATE-01-01). We thank Audi.JKU Deep Learning Center, TGW LOGISTICS GROUP GMBH, Silicon Austria Labs (SAL), University SAL Labs initiative, FILL Gesellschaft mbH, Anyline GmbH, Google, ZF Friedrichshafen AG, Robert Bosch GmbH, UCB Biopharma SRL, Merck Healthcare KGaA, Verbund AG, GLS (Univ. Waterloo) Software Competence Center Hagenberg GmbH, TÜV Austria, Frauscher Sensonic, TRUMPF and the NVIDIA Corporation.
We acknowledge EuroHPC Joint Undertaking for awarding us access to Karolina at IT4Innovations (Czech Republic) and MeluXina at LuxProvide (Luxembourg).

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

# A Environments

## A.1 Meta-World & Continual-World

**Meta-World**. For our experiments, we use the Meta-World benchmark (Yu et al., 2020b). Meta-World tasks consist of a Sawyer robotic arm simulated using the MuJoCo physics engine (Todorov et al., 2012). The observation space in Meta-World is a 39-dimensional vector. The action space is 4-dimensional, with all actions in range $[-1, 1]$. For each task, the reward functions are scaled to have a maximum value of 10 and a minimum of 0. The exact reward-function definitions are provided in Yu et al. (2020b). Episodes last for 200 timesteps for each task. For all our experiments on Meta-World, we report success rates and mean rewards obtained.

**Difference between Meta-World v1 and v2 environments.** The Continual-World benchmark was published using v1 of Meta-World. However, in the meantime v2 has been released. While the v1 environments had a 9 dimensional observation space, the observation space is 39-dimensional in v2. Another major change in v2, was the introduction of dense and bounded reward functions.

**Continual-World.** The Continual-World benchmark was proposed by Wolczyk et al. (2021) and is built on top of Meta-World. Continual-World is a challenging CRL benchmark, consisting of 10 of the 50 tasks contained in Meta-World, denoted as CW10. We use the tasks contained in CW10 as fine-tuning set and select the remaining 40 tasks as pre-training set (MT40). The CW10 task sequence is:

*hammer-v2*, *push-wall-v2*, *faucet-close-v2*, *push-back-v2*, *stick-pull-v2*, *stick-pull-v2*, *handle-press-side-v2*, *push-v2*, *shelf-place-v2*, *window-close-v2*, and *peg-unplug-side-v2*.

## A.2 DeepMind Control Suite (DMControl)

The DMControl Suite benchmark (Tassa et al., 2018) consists of 30 simulated continuous control tasks built on the MuJoCo physics engine. DMControl encompasses a wide variety of robot morphologies, ranging from a simple cartpole to a complex humanoid. Therefore, the observation and action spaces vary across environments. The action spaces vary between 1 (`cartpole`) and 21 continuous dimensions (`humanoid`), where each action dimension is bounded by $[-1, 1]$. Agents can be trained based on continuous vector-based representations (state-based) or directly from pixels (pixel-based). In this work, we focus on the state-based representations, which vary between 3 (`pendulum`) and 67 continuous dimensions (`humanoid`). As described in Section 3, we select 16 of the 30 DMControl tasks and assign 10 to the pre-training set (DMC10) and 6 to the fine-tuning set (DMC6) (similar to Hafner et al. (2019)). We list the dimensionalities of their state and action spaces in Table 4.

The **DMC10** tasks include:

*cartpole-balance*, *finger-turn_easy*, *finger-turn_hard*, *fish-upright*, *hopper-stand*, *pendulum-swingup*, *point_mass-easy*, *reacher-hard*, *walker-run*, *walker-stand*

Furthermore, the **DMC6** tasks include:

*ball_in_cup-catch*, *cartpole-swingup*, *cheetah-run*, *finger-spin*, *reacher-easy*, *walker-walk*

For all environments in DMControl, episodes last for 1000 timesteps and the maximum achievable return is 1000. In addition to the mean reward obtained, we report data-normalized scores, as suggested by Fu et al. (2020). For each environment, the scores are normalized by the average scores achieved by the expert agent used to collect the dataset, and the performance attained by a random agent, $\texttt{normalized\_score} = \frac{\texttt{score}-\texttt{random\_score}}{\texttt{expert\_score}-\texttt{random\_score}}$. Therefore, the normalized scores are roughly between 0 and 1. The expert and random scores used for score normalization are available in the accompanying source code[2].

# B Methods

**Fine-tuning.** In full fine-tuning, the entire pre-trained model is trained on the new task. We also try other common variations thereof, such as fine-tuning the action head, the last layer, or both.

---

[2]Available at: `https://github.com/ml-jku/L2M`

**Adapters.** Adapters are a parameter-efficient fine-tuning approach proposed by Houlsby et al. (2019). An adapter layer consists of a down-projection, a non-linearity and an up-projection along with a skip connection. Each Transformer block is interleaved with two Adapter modules. During training, only the Adapters (and optionally LayerNorms) are updated, while the remaining parameters remain frozen. This significantly reduces the number of trainable parameters, while preserving the ability to adapt to new tasks. Houlsby et al. (2019) adapt a BERT (Devlin et al., 2019) model to 26 different text classification tasks, while training only roughly 4% of the parameters and attaining performance close to full fine-tuning.

**(IA)$^3$: Infused Adapter by Inhibiting and Amplifying Inner Activations.** Similar to LoRA, (IA)$^3$ (Liu et al., 2022b) modulates the inner activation flow of the Transformer layers and can be applied to every component of the pre-trained model. Instead of low-rank modulation matrices as used in LoRA, (IA)$^3$ performs elementwise multiplication with learnable modulation vectors. Thus, (IA)$^3$ typically incurs less additional parameters compared to other methods.

**Prompt-tuning.** Lester et al. (2021) learn soft prompts to enable a frozen pre-trained GPT-3 (Brown et al., 2020) model to generalize to downstream tasks. In prompt-tuning, the learnable prompts are prepended to the input sequence, while the rest of the pre-trained model is kept frozen.

**Prefix-tuning.** Prefix-tuning (Li and Liang, 2021) is another PBT-approach. Similar to prompt-tuning, prefix-tuning prepends learnable vectors (i.e., soft-prompts) to the input. However, unlike in prompt-tuning, the learnable prompts are prepended to the keys and values of the attention head input. Therefore, prefix-tuning also incurs more parameters than prompt-tuning, but typically results in better performance.

**P-tuning v2.** Liu et al. (2021b) proposed P-tuning v2 as a successor of prefix-tuning. Instead of prepending continuous prompts only on the input layer, P-tuning v2 prepends them for every layer of the pre-trained model. This simple modification results in more parameters, but better performance. In particular, P-tuning v2 matches the performance of full fine-tuning on a number of NLP benchmarks and is effective across model sizes (330M to 10B parameters).

**EWC and L2.** EWC (Kirkpatrick et al., 2017) is an established technique for continual learning based on regularization. EWC helps to prevent forgetting of previous tasks when learning a new task by protecting parameters that are important for previously learned tasks. EWC uses the Fisher information matrix as a regularization term, which measures the sensitivity of each parameter with respect to each task and, thus, indicates which parameters require protection. L2 is used as a baseline in the original EWC publication (Kirkpatrick et al., 2017) and utilizes the L2 penalty to protect previously learned weights from being changed.

## C  Multi-Domain Decision Transformer (MDDT)

In this section, we describe our Multi-Domain Decision Transformer (MDDT) architecture in more detail. We pre-train a MDDT on the 50 collected pre-training datasets simultaneously (Section 3.1). Then we fine-tune the pre-trained model to 16 new tasks from two domains. To achieve this, we extend the DT architecture proposed by Chen et al. (2021) to handle inputs from multiple domains with varying state and/or action spaces.

**Unified state-space.** While all Meta-World tasks share a 39-dimensional state-space, the dimensionalities of the state-spaces in DMControl vary between 3 and 67 continuous values (See Appendix A). This variation reflects the diverse robot morphologies across DMControl tasks. To deal with these heterogeneous state spaces, we construct a unified state-space that spans all dimension of DMControl and Meta-World tasks. While the dimensionalities of state-spaces differ between tasks, they may share common observation types, such as orientation, current velocity, or arm positions of the robot. In total, DMControl encompasses 24 different observation types. Our constructed state-space unifies all of these observation-type attributes, amounting to 204 continuous dimensions in total. For example, the first 13 dimensions correspond to the current orientations of the robot and the last dimension corresponds to the current height. Dimensions that are unused by a certain task are zero-padded. Thus, the constructed state-space covers all environments considered in this work. Finally, as in the original DT publication (Chen et al., 2021), the states are embedded with a linear layer before serving as input to the MDDT.

We choose this scheme for handling varying state-spaces instead of discretisation followed by padding, as was used in prior work (Reed et al., 2022). Discretisation of states in combination with padding drastically increases the context length and, thus, the computational requirements. For example, when tokenising a trajectory of five timesteps in Meta-World, the context would expand to 195 tokens (39 dimensions * 5 timesteps) for states alone. In contrast, our scheme results in only 5 state tokens.

**Action tokenisation & autoregressive prediction.** To handle action spaces of varying dimensionalities, we tokenize each action dimension and autoregressively predict action tokens, similar to Reed et al. (2022) and Brohan et al. (2022). In the environments we consider, the action-spaces vary between 1 and 6 dimensions and all actions are bounded by [-1, 1]. We use a min-max tokenisation to discretise every continuous action dimensions into 64 bins and then pad to the maximum action dimension. Thus, we obtain 6 discrete action tokens from every action. During evaluation, we autoregressively predict the action tokens until the dimensionality of action space in the particular task is reached, and then execute the composed action. Note that autoregressive prediction of actions make the prediction task harder, compared to simply predicting the ground truth action, as is commonly done for continuous control tasks.

**Context representation.** Similar to Lee et al. (2022), the policy $\pi_\theta$ with trainable parameters $\theta$, predicts the next action $a_t$ from a given context of the $K$ most recent states, RTGs, actions and rewards. States and actions are pre-processed as described earlier, and then embedded using separate linear layers. Similarly, we embed the continuous scalars for RTGs and rewards using linear layers. Unlike Lee et al. (2022), we do not discretise rewards or RTGs. However, similar to Chen et al. (2021), we scale the rewards and RTGs such that all RTGs lie roughly within the range [0, 10]. To achieve this, we use separate reward scales of 200 and 100, for Meta-World and DMControl, respectively. Thus, each timestep is made up of 9 embedded tokens ($1\times$ state, $1\times$ RTG, $6\times$ action, $1\times$ reward). We use a context-length of 5 timesteps for all our experiments, and therefore, the final sequence length expands to 45 embedded tokens (5 timesteps * 9 tokens per timestep).

**Positional encodings.** Once the sequence is embedded, we add positional information to every embedded token. As Chen et al. (2021), we learn an embedding for each timestep and add this embedding to every embedded token (i.e., states/action/reward/rtg tokens) in a timestep. This provides the DT with vital information about the absolute position within the current episode. This is important since the choice for an action may differ at the beginning vs. at the end of an episode. We also explored other positional encodings in preliminary experiments, but found learned time-step based embeddings to work best.

**Training & Evaluation.** In line with prior work (Chen et al., 2021; Lee et al., 2022), we train the DT via return-conditioned upside-down RL using a cross-entropy loss to predict next actions. During evaluation, we set the target return to the maximum observed return in the respective dataset, instead of utilizing the expert-action inference mechanism proposed by Lee et al. (2022). However, we also found that constant proxies per domain (e.g., 2000 and 1000) work well.

These simple yet effective modifications enable processing observations and actions that originate from different environments with varying state and/or action spaces at training and test time. However, a few notable **limitations** remain:

- The constructed **state-space** is specific to the environments we consider in our work and, thus, not directly transferable to other environments. Therefore, a new unified state-space may have to be constructed (same procedure), when using a different environment.
- The **discretised actions** are padded to the maximum action dimension. Therefore, an input sequence for an environment with a smaller than the maximum action dimension, may contain multiple redundant tokens. However, this design decision simplified our implementation considerably.
- During evaluation, we use **autoregressive prediction** of action dimensions. This makes the prediction task considerably harder, when compared to predicting all dimensions at once. Thus, this can result in lower overall performance (see Appendix 23).

# D   Dataset

To collect our dataset, we train expert agents via SAC (Haarnoja et al., 2018). We train a separate expert on each of the 66 considered tasks.

**Table 2:** Performance scores for data collection on MT40.

| Task | $|\mathcal{S}|$ | $|\mathcal{A}|$ | Success Rate | Reward |
|---|---|---|---|---|
| assembly-v2 | 39 | 4 | 0.0 | 1206.9 |
| basketball-v2 | 39 | 4 | 0.9 | 1375.95 |
| bin-picking-v2 | 39 | 4 | 0.0 | 474.81 |
| box-close-v2 | 39 | 4 | 0.0 | 759.15 |
| button-press-topdown-v2 | 39 | 4 | 1.0 | 1299.24 |
| button-press-topdown-wall-v2 | 39 | 4 | 1.0 | 1296.16 |
| button-press-v2 | 39 | 4 | 1.0 | 1430.44 |
| button-press-wall-v2 | 39 | 4 | 1.0 | 1508.16 |
| coffee-button-v2 | 39 | 4 | 1.0 | 1499.17 |
| coffee-pull-v2 | 39 | 4 | 1.0 | 1313.88 |
| coffee-push-v2 | 39 | 4 | 0.6 | 508.14 |
| dial-turn-v2 | 39 | 4 | 0.8 | 1674.29 |
| disassemble-v2 | 39 | 4 | 1.0 | 1396.55 |
| door-close-v2 | 39 | 4 | 1.0 | 1535.4 |
| door-lock-v2 | 39 | 4 | 1.0 | 1712.65 |
| door-open-v2 | 39 | 4 | 1.0 | 1544.32 |
| door-unlock-v2 | 39 | 4 | 1.0 | 1733.64 |
| drawer-close-v2 | 39 | 4 | 1.0 | 1845.92 |
| drawer-open-v2 | 39 | 4 | 1.0 | 1710.65 |
| faucet-open-v2 | 39 | 4 | 0.9 | 1727.98 |
| hand-insert-v2 | 39 | 4 | 1.0 | 1607.17 |
| handle-press-v2 | 39 | 4 | 1.0 | 1854.79 |
| handle-pull-side-v2 | 39 | 4 | 1.0 | 1613.72 |
| handle-pull-v2 | 39 | 4 | 1.0 | 1581.75 |
| lever-pull-v2 | 39 | 4 | 1.0 | 1449.05 |
| peg-insert-side-v2 | 39 | 4 | 1.0 | 1545.19 |
| pick-out-of-hole-v2 | 39 | 4 | 1.0 | 1435.64 |
| pick-place-v2 | 39 | 4 | 0.0 | 6.59 |
| pick-place-wall-v2 | 39 | 4 | 0.1 | 702.59 |
| plate-slide-back-side-v2 | 39 | 4 | 1.0 | 1766.24 |
| plate-slide-back-v2 | 39 | 4 | 1.0 | 1773.56 |
| plate-slide-side-v2 | 39 | 4 | 1.0 | 1663.35 |
| plate-slide-v2 | 39 | 4 | 1.0 | 1667.35 |
| reach-v2 | 39 | 4 | 1.0 | 1858.99 |
| reach-wall-v2 | 39 | 4 | 1.0 | 1831.14 |
| soccer-v2 | 39 | 4 | 0.4 | 445.84 |
| stick-push-v2 | 39 | 4 | 1.0 | 1470.71 |
| sweep-into-v2 | 39 | 4 | 1.0 | 1761.69 |
| sweep-v2 | 39 | 4 | 1.0 | 1458.35 |
| window-open-v2 | 39 | 4 | 1.0 | 1537.59 |
| Average | - | - | $0.84 \pm 0.34$ | $1414.62 \pm 439.39$ |

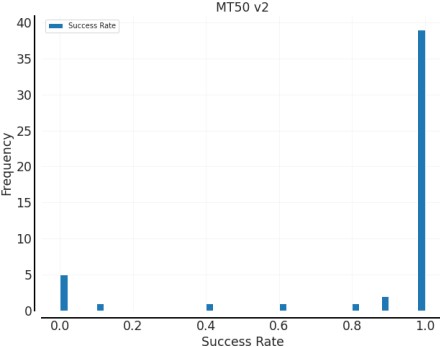

**Figure 9:** Histogram for success rates over all tasks. The majority of all MT50 tasks can be solved successfully at the end of training by single-task SAC, while on a minority of tasks the agents fails to achieve successful completion.

**Table 3:** Performance scores for data collection on CW10.

| Task | $|\mathcal{S}|$ | $|\mathcal{A}|$ | Success Rate | Reward |
|------|------|------|------|------|
| faucet-close-v2 | 39 | 4 | 1.0 | 1768.87 |
| hammer-v2 | 39 | 4 | 1.0 | 1632.21 |
| handle-press-side-v2 | 39 | 4 | 1.0 | 1842.17 |
| peg-unplug-side-v2 | 39 | 4 | 1.0 | 1373.45 |
| push-back-v2 | 39 | 4 | 1.0 | 1373.32 |
| push-v2 | 39 | 4 | 1.0 | 1672.88 |
| push-wall-v2 | 39 | 4 | 1.0 | 1594.37 |
| shelf-place-v2 | 39 | 4 | 1.0 | 1376.92 |
| stick-pull-v2 | 39 | 4 | 1.0 | 1344.29 |
| window-close-v2 | 39 | 4 | 1.0 | 1426.45 |
| Average | 39 | 4 | $1.0 \pm 0.0$ | $1540.49 \pm 184.43$ |

### D.1 Meta-World

For each of the 50 Meta-World tasks, we train for 2M steps and record the entire replay buffer. Thus, the final Meta-World datasets contain 2M transitions (i.e., state-RTG-action-reward tuples). Each trajectory is 200 steps long, and each dataset consists of 10K trajectories. This amounts to 100M transitions in total for all 50 tasks, 80M and 20M for MT40 and CW10, respectively.

We use the same network architecture as Wolczyk et al. (2021), 4 linear layers with 256 neurons, LayerNorm (Ba et al., 2016) after the first layer, and LeakyReLU (Maas et al., 2013) activations for both, actor and critic. We use $\alpha = 0.01$ for the LeakyReLU instead of $\alpha = 0.2$ used by Wolczyk et al. (2021), since it lead to performance improvements.

After every 50 interaction steps, we perform 50 gradient steps using Adam (Kingma and Ba, 2015). We keep other parameters fixed at their default values in `stable-baselines3` (Raffin et al., 2021). This includes the learning rate of $3e^{-4}$, batch size of 256, discount factor 0.99, and automatic entropy tuning. The target networks are synced after every update step with an update coefficient of 0.005. We evaluate the current policy after every 10K interaction steps. Each evaluation run consists of 10 individual evaluation episodes, and scores are averaged over all 10 episodes. The success rates provide a measure of trajectory quality (TQ, Schweighofer et al., 2022) for all individual datasets. Additional measures such as state-action coverage (SACo) may also be computed.

In Tables 2 and 3, we list the success rates and average rewards achieved by the expert agents on MT40 and CW10 tasks, respectively. In addition, we show the success rate distribution in Figure 9 and the learning curves on MT40 and CW10 in Figures 10 and 11, respectively. Across tasks, we observe notable differences in learning behaviour. The expert learns some tasks already after a few thousand interaction steps, while it requires more interaction steps for others. For some it does not

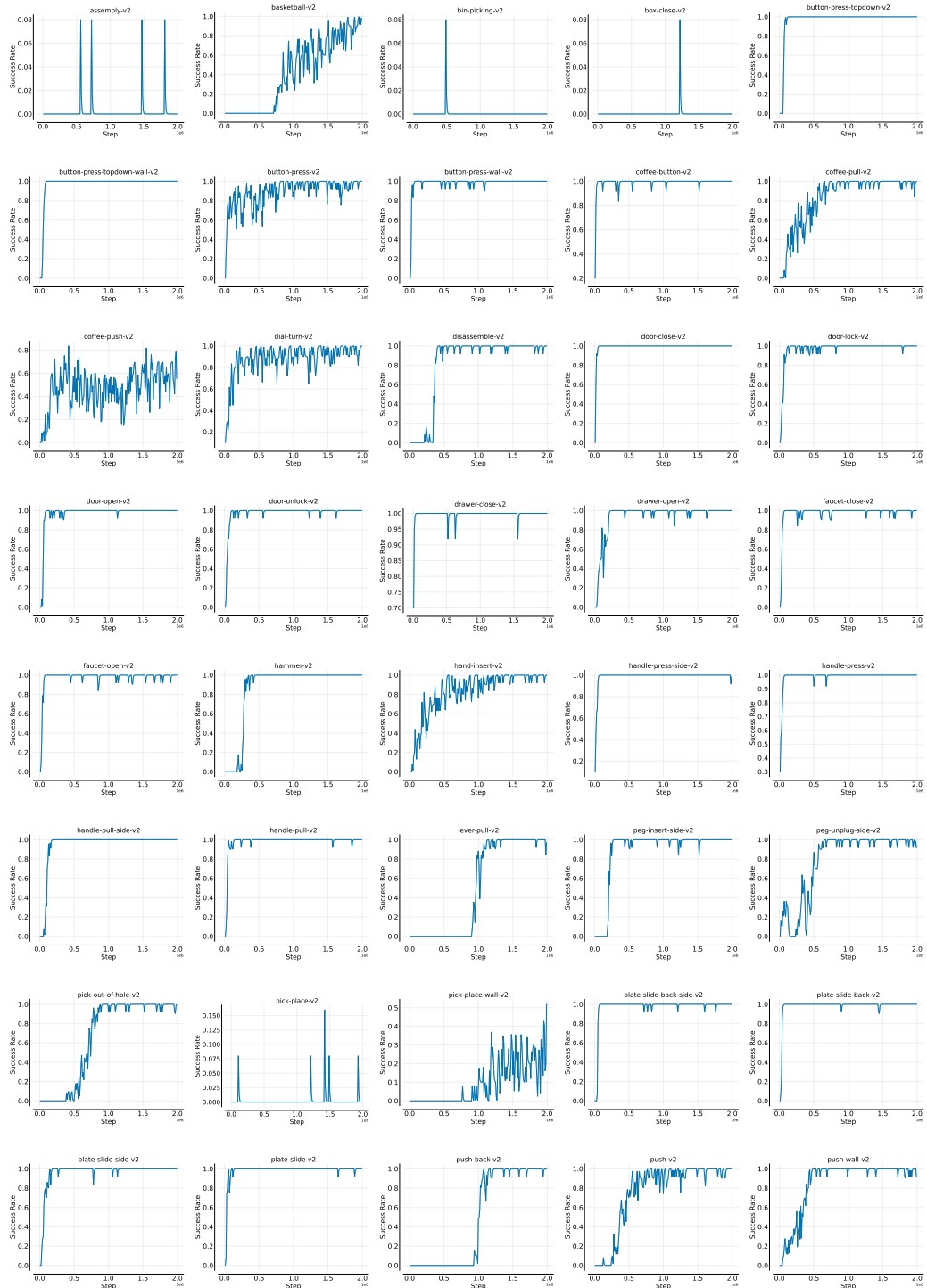

**Figure 10:** Learning curves for data-collection runs on all MT40 tasks with SAC. We train for 2M environment interaction steps on each task and record the entire replay buffer.

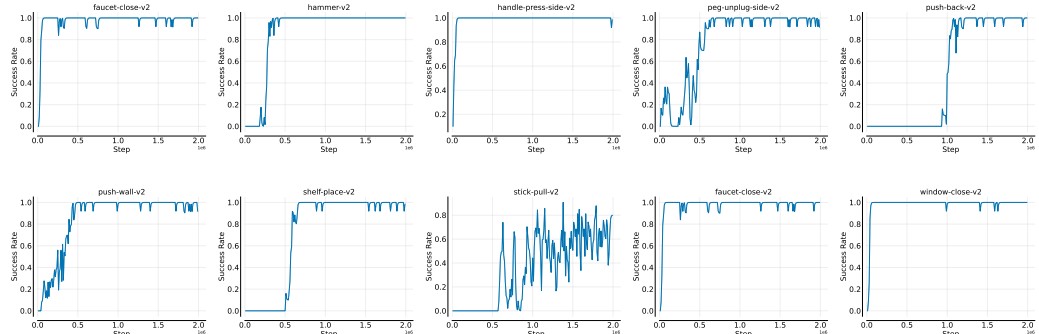

**Figure 11:** Learning curves for data-collection runs on all CW10 tasks with SAC. We train for 2M environment interaction steps on each task and record the entire replay buffer.

exhibit any learning progress at all (e.g. *assembly-v2*). Overall, the expert achieves average success rates of 84% and 100% on MT40 and CW10, respectively.

### D.2 DMControl

For each of the 16 considered DMControl tasks, we train for 1M steps. Therefore, the final DMControl datasets contain 1M transitions per task, amounting to 16M transitions in total (10M for DMC10, 6M for DMC6). All trajectories are of the same length and contain 1000 timesteps.

We utilize the same network architecture as for Meta-World (Section D.1), using 1024 neurons per layer instead of 256. Furthermore, we perform one gradient step per interaction step and keep all other hyper-parameters the same as on Meta-World.

We list the average rewards achieved by the expert agents on DMC10 and DMC6 tasks in Table 4. The corresponding learning curves for all DMC10 tasks are available in Figure 12. On both splits, the agent performs close to the best possible return of 1000, achieving a mean reward of 788 on DMC10 and 840 on DMC6.

**Table 4:** Performance scores for data collection on **(a)** DMC10 and **(b)** DMC6.

(a) DMC10

| Task | $|\mathcal{S}|$ | $|\mathcal{A}|$ | Reward |
|---|---|---|---|
| cartpole-balance | 5 | 1 | 967.49 |
| finger-turn_easy | 12 | 2 | 940.57 |
| finger-turn_hard | 12 | 2 | 940.15 |
| fish-upright | 21 | 5 | 787.87 |
| hopper-stand | 15 | 4 | 417.49 |
| pendulum-swingup | 3 | 1 | 682.85 |
| point_mass-easy | 4 | 2 | 842.03 |
| reacher-hard | 6 | 2 | 945.18 |
| walker-run | 24 | 6 | 402.36 |
| walker-stand | 24 | 6 | 957.61 |
| Average | - | - | 788 ± 219 |

(b) DMC6

| Task | $|\mathcal{S}|$ | $|\mathcal{A}|$ | Reward |
|---|---|---|---|
| ball_in_cup-catch | 8 | 2 | 971.1 |
| cartpole-swingup | 5 | 1 | 823.48 |
| cheetah-run | 17 | 6 | 411.76 |
| finger-spin | 9 | 2 | 961.85 |
| reacher-easy | 6 | 2 | 890.03 |
| walker-walk | 24 | 6 | 928.84 |
| Average | - | - | 840 ± 216 |

## E  Pre-training

We train our MDDT for a total of 1M update steps, with context length of 5 transitions (45 tokens). We use a learning rate of $1e^{-4}$ and 4000 linear warm-up steps, followed by a cosine decay to $1e^{-6}$. Furthermore, we use gradient clip of 0.25, weight decay of 0.01, dropout of 0.2, a batch size of

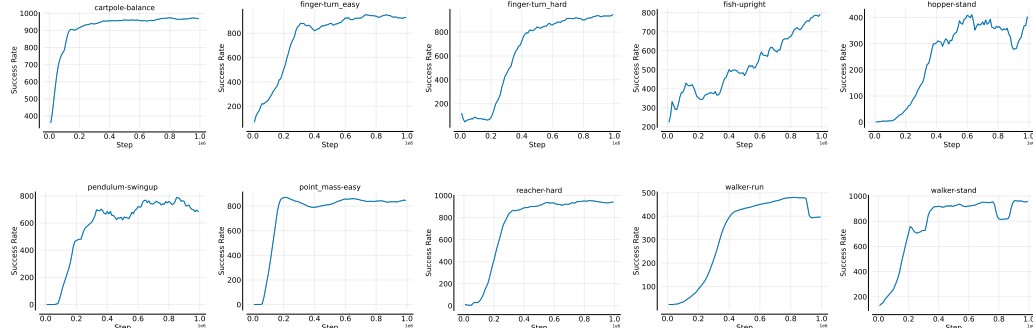

**Figure 12:** Learning curves for data-collection runs on all DMC10 tasks with SAC. We train for 1M environment interaction steps on each task and record the entire replay buffer.

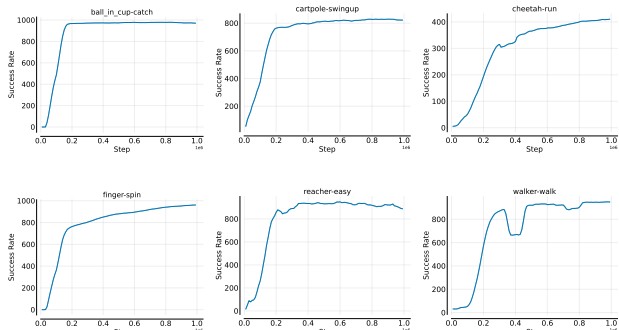

**Figure 13:** Learning curves for data-collection runs on all DMC6 tasks with SAC. We train for 1M environment interaction steps on each task and record the entire replay buffer.

1024 sequences and train using the AdamW optimizer (Loshchilov and Hutter, 2018). We base our implementation on the DT implementation in the `transformers` library (Wolf et al., 2020), and keep their default values for remaining parameters.

We use the same underlying GPT-2 like network architecture as Lee et al. (2022). However, we do not use their proposed expert-action inference mechanism. Instead, we set the target return to the maximum return in the respective dataset and use a reward scale of 200 and 100 for Meta-World and DMControl, respectively. Furthermore, we use timestep-based positional embeddings, as described in Appendix C.

In Table 5 and Figure 14, we show the aggregate scores over all tasks at the end of training, and learning curves across different model sizes, respectively. We vary the number of layers, the number of heads, and the embedding dimension. We also experimented with higher context lengths, but only found slight performance gains at higher computational cost. Due to the excessive computational training cost, we only train one model per size. Generally, there is a trend that more complex models

**Table 5:** Pre-training scores for multi-domain DT variants (40M-200M) trained on MT40 and DMC10. We vary the number of layers, the number of heads, and the embedding dimension. We report success rates for MT40, normalized scores for DMC10, and mean rewards across both domains.

| Layers | Heads | Embedding Dim | Params | MT40 | DMC10 | Mean Reward |
|--------|-------|---------------|--------|------|-------|-------------|
| 6 | 12 | 768 | 40M | 0.6 | 0.84 | 983.92 |
| 8 | 12 | 768 | 58M | 0.6 | 0.93 | 1005.6 |
| 8 | 16 | 1024 | 102M | 0.61 | 1.21 | 1055.75 |
| 10 | 20 | 1280 | 200M | 0.6 | 1.33 | 1030.1 |

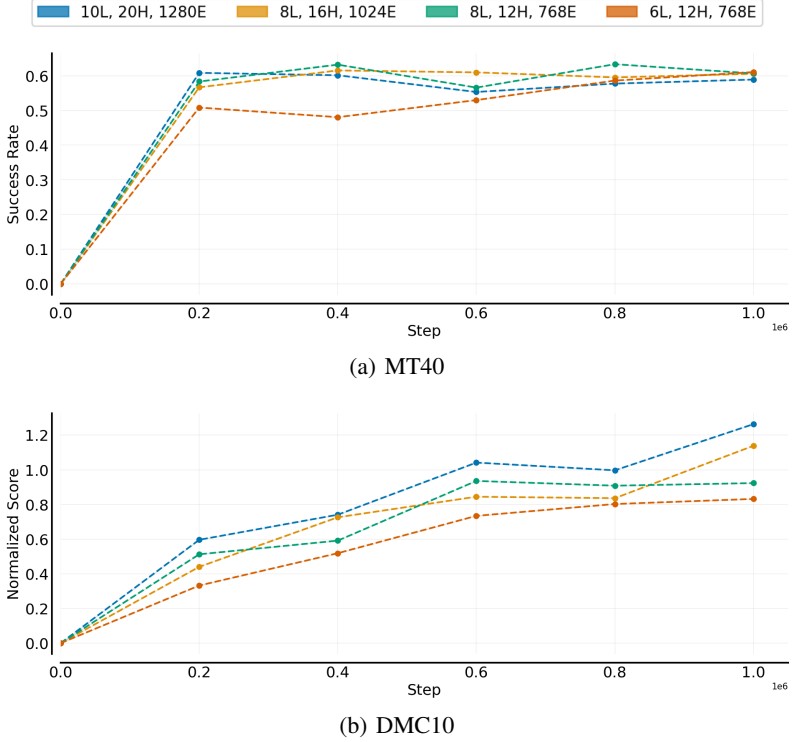

(a) MT40

(b) DMC10

**Figure 14:** Learning curves for multi-domain pre-training runs on MT40 and DMC10 with different architectural choices of the Transformer network (L = # of layers, H = # of heads, E = embedding dimension). We train every model variant for 1M update steps with a batch size of 1024.

learn faster and attain higher performance on DMC10. However, for all our subsequent experiments, we used the model with 40M parameters, as it achieves a good trade-off between performance and training time. After pre-training, we evaluate this model across 3 environment seeds and attain an average success rate of 60% ($\pm 0.5\%$) on MT40 and a normalized score of 94% ($\pm 11\%$) on DMC10.

## F  Task Separation Analysis

To better understand what the DT learned, we visualize token embeddings using t-SNE (Van der Maaten and Hinton, 2008). First, we sample 256 state-RTG-action-reward sequences of length 5 (i.e., 45 tokens) for each task and record the hidden states of each (frozen) Transformer block. For each sequence, we then average the hidden states over the sequence, either across the entire sequence or individually per token-type (i.e., for states, RTGs, actions, rewards). We use the 40M pre-trained MDDT for our analysis. Thus, we end up with 256 768-dimensional vectors per task. We cluster all vectors using t-SNE projecting into a two-dimensional space with a perplexity of 30, 10K optimization steps, and the cosine distance.

In Figure 15, we show the t-SNE visualizations of all tokens combined, as well as for state, action, RTG, and reward tokens individually on 10 tasks in MT40 and DMC10. For embedded state tokens, we observe good cluster separation. Also, for all tokens combined and actions some clusters can be observed, but no task-specific separation. For rewards and RTG, we do not observe distinct clusters. This is expected, since rewards and RTGs do not vary substantially among tasks.

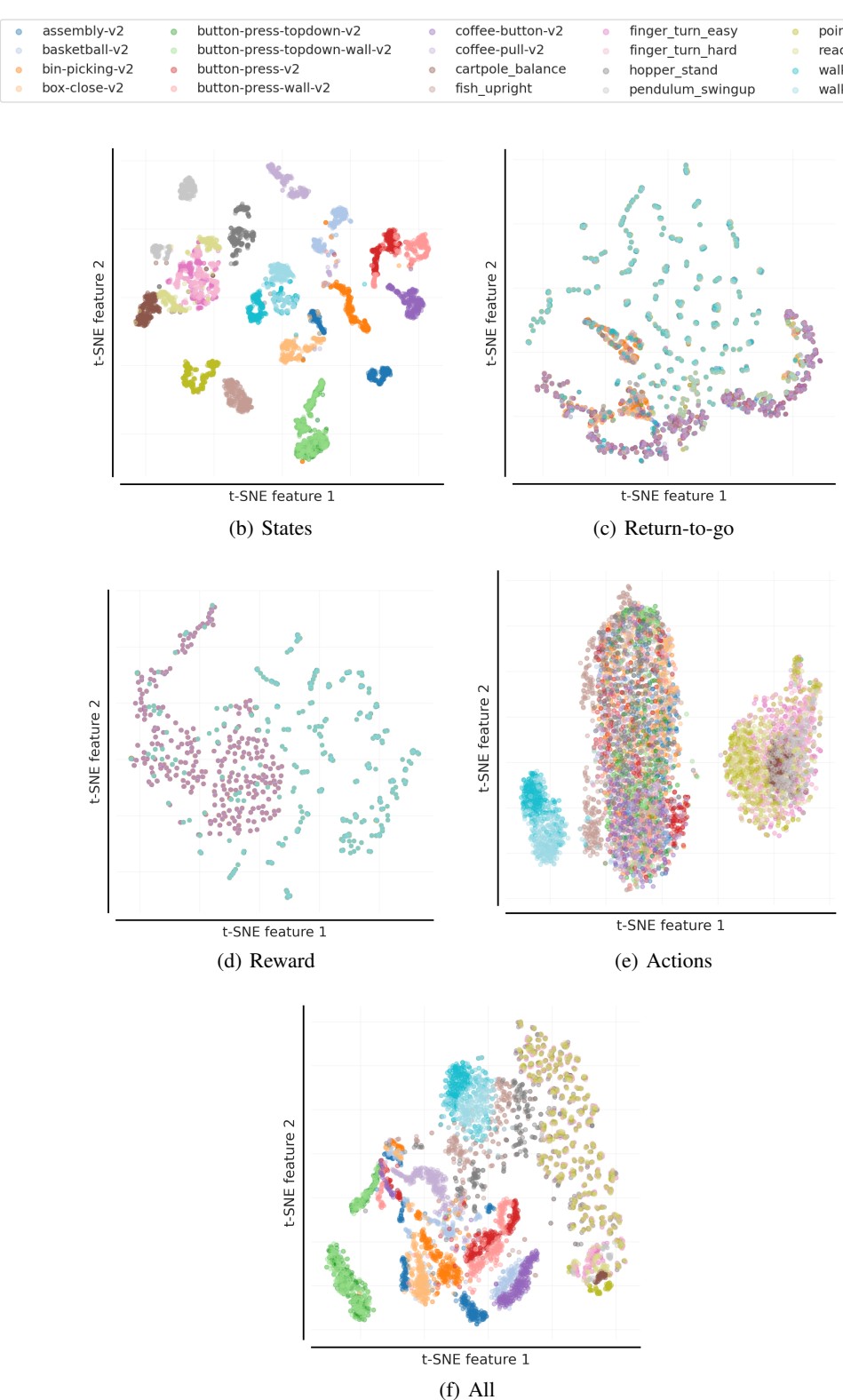

**Figure 15:** t-SNE clustering for state, action, RTG, and reward embeddings individually, as well as for all token embeddings combined for the first ten tasks in MT40 and DMC10.

**Table 6:** Aggregate scores for all single-task fine-tuning experiments on CW10 and DMC6 datasets.

| Method | CW10 | | DMC6 | |
| --- | --- | --- | --- | --- |
| | **Success Rate** | **Mean Reward** | **Norm. Score** | **Mean Reward** |
| FT | 0.83 ± 0.21 | 1449.15 ± 329.56 | 1.14 ± 0.09 | 832.46 ± 198.22 |
| FT-head+last | 0.7 ± 0.24 | 1232.9 ± 422.91 | 0.69 ± 0.37 | 547.76 ± 324.0 |
| FT-head | 0.03 ± 0.11 | 127.23 ± 205.68 | 0.08 ± 0.09 | 81.23 ± 76.37 |
| Adapters | 0.76 ± 0.23 | 1330.29 ± 346.19 | 0.85 ± 0.23 | 662.63 ± 255.08 |
| LoRA | 0.75 ± 0.24 | 1319.89 ± 365.68 | 0.79 ± 0.26 | 616.8 ± 276.89 |
| $(IA)^3$ | 0.64 ± 0.35 | 1131.7 ± 436.49 | 0.23 ± 0.26 | 195.95 ± 195.35 |
| P-tuning v2 | 0.31 ± 0.39 | 556.97 ± 571.94 | 0.11 ± 0.11 | 102.33 ± 94.55 |
| Prefix-tuning | 0.12 ± 0.25 | 282.6 ± 468.18 | 0.12 ± 0.14 | 112.24 ± 117.92 |
| Prompt-tuning | 0.09 ± 0.23 | 178.25 ± 373.56 | 0.07 ± 0.08 | 76.27 ± 74.87 |
| FT-MT-scratch | 0.85 ± 0.07 | 1476.61 ± 395.13 | 0.92 ± 0.03 | 648.49 ± 354.7 |
| FT-MT-pre-trained | 0.92 ± 0.02 | 1491.76 ± 332.0 | 1.04 ± 0.17 | 764.38 ± 230.28 |

# G  Single-Task Fine-Tuning

We conduct a broad evaluation of different FT, PEFT and PBT methods to adapt the MDDT to each of the held-out fine-tuning tasks in a single-task setting (Section 3.2). In this setting, we present results for fine-tuning on all CW10 and DMC6 tasks individually.

We compare 9 different methods on CW10 and DMC6, as listed in Section 3.2. We train for 100K update steps on each task. Every 10K update steps, we evaluate the current model within the actual environment for 10 evaluation episodes and average performance across the evaluation episodes. The final performance scores are aggregated over all tasks in the task sequence. We show the aggregated task performances on CW10 and DMC6 in Table 6. In addition, we provide the respective learning curves in Figure 16. In Figure 17 we compare the fraction of trainable parameters vs. attained performance for all considered fine-tuning techniques on DMC6.

For FT variations, Adapters, LoRA and $(IA)^3$, we use a learning rate of $1e^{-4}$. For all PBT methods, we increase the learning rate to $1e^{-3}$ which resulted in better performance. For Adapters, we set a reduction factor of 16 for the down-projection. For LoRA, we set the rank to 8 by default, but conduct an ablation on this choice in Section G.1. For all prompt-tuning approaches, we use a prompt length of 25 and a dropout rate of 0.2. We found these values to work well in preliminary experiments. For LoRA, Adapters and $(IA)^3$ we do not use Dropout (Srivastava et al., 2014).

Moreover, we add the performance scores for two multi-task baselines in Table 6. While the first one is trained from scratch (FT-MT-scratch), the second is initialised from the pre-trained model (FT-MT-pre-trained). Both model variants are trained for 1M update steps on the multi-domain mix of CW10 and DMC6. The rationale for this comparison, is to investigate the effect of pre-training when learning new tasks. Indeed, we observe significantly higher fine-tuning scores on both domains for the pre-trained model.

## G.1  What rank is required in LoRA?

To investigate the impact of the rank $r$ in LoRA, we performed a hyper-parameter search varying $r$ across the values $\{1, 2, 4, 8, 16, 32\}$. We present the results of our experiments on CW10 and DMC6 in Figure 18. Specifically, we plot the performance against the percentage of parameters trained. Note that the x-axis is on a log-scale. For all experiments presented in Sections 3.2 and 3.3, we used a rank of 8 for LoRA. As expected, increasing the rank can improve performance on both considered domains. However, this improvement comes at the expense of more trainable parameters.

## G.2  What matrices should be modulated using LoRA?

Another important design choice when using LoRA is which of the weights of the pre-trained model to modulate. To investigate this, we conduct an ablation study in which we vary the modulation targets. In principle, LoRA can be applied to any weight matrix in the pre-trained model. In our

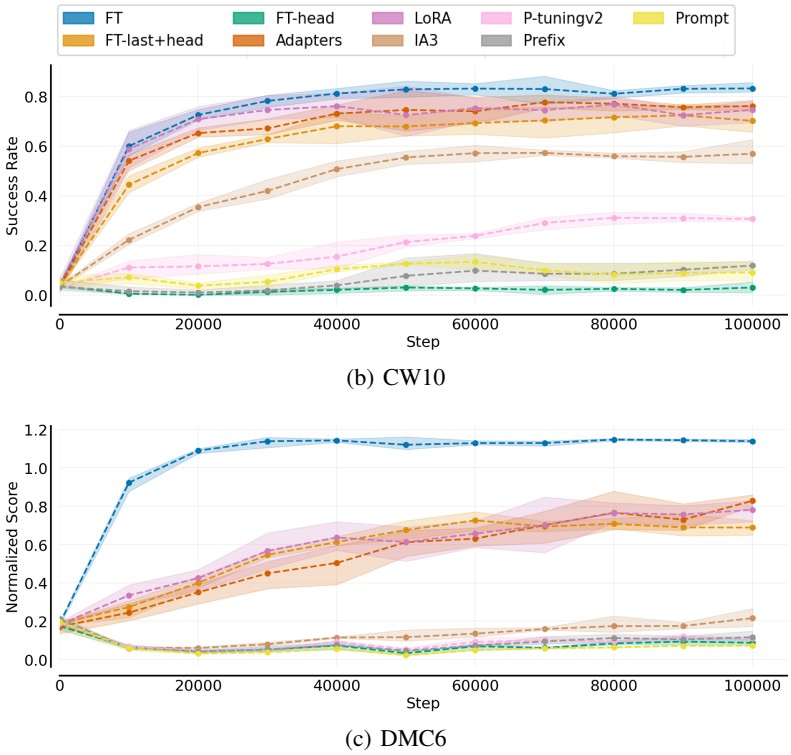

(b) CW10

(c) DMC6

**Figure 16:** Learning curves for single-task fine-tuning experiments on **(a)** CW10 and **(b)** DMC6. On every task, we train for 100K timesteps and consequently average the performances across all tasks in the data split.

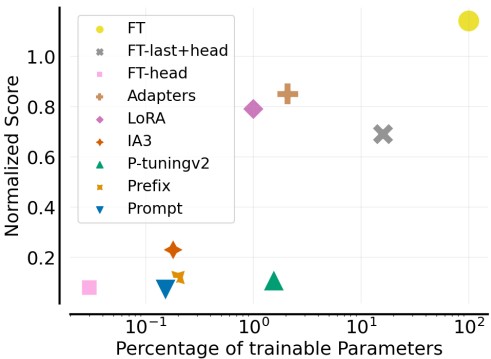

**Figure 17:** Normalized score vs. fraction of parameters trained for various fine-tuning techniques on single-task experiments for DMC6.

ablation study, we specifically focus on modulating the keys, values, and queries in the self-attention mechanism, as well as the position-wise feedforward layer.

We present the results of our experiments in Figure 19. We find that modulating the position-wise feedforward layer tends to be more important the modulating the self-attention mechanism. Overall, the best performance is obtained when modulating all considered targets. However, this comes at the cost of more parameters. Notably, these results narrow the performance gap to L2M-oracle.

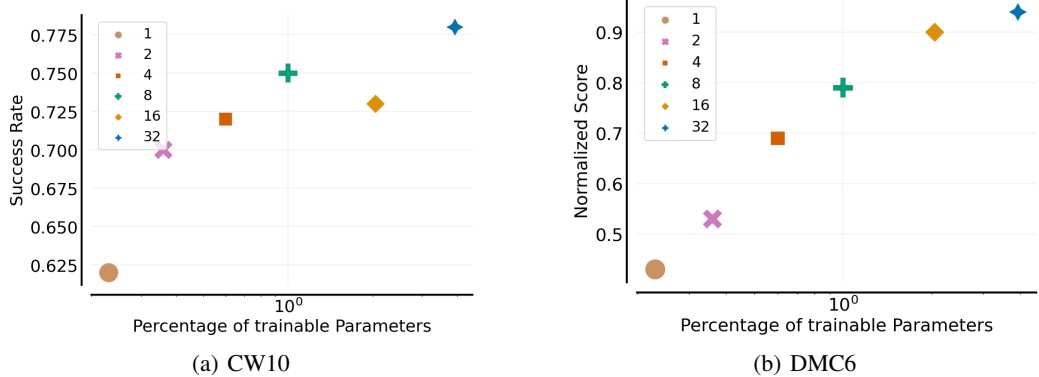

(a) CW10          (b) DMC6

**Figure 18:** Performance vs. fraction of parameters trained for different values for the rank in LoRA on **(a)** CW10 and **(b)** DMC6.

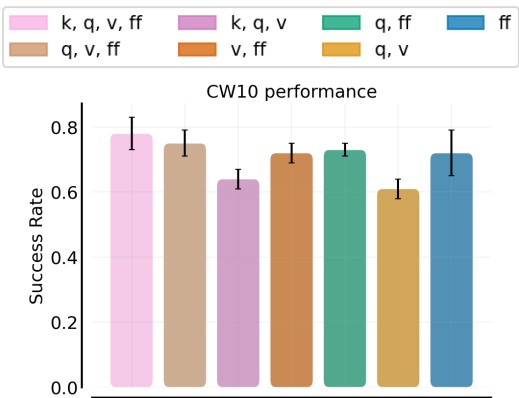

**Figure 19:** Modulation target ablation for L2M on CW10.

## H    Continual Fine-Tuning

Ultimately, our goal is to adapt the pre-trained model to multiple novel tasks, while alleviating forgetting of tasks learned during pre-training. In this regard, we adapt the pre-trained model to all fine-tuning tasks in a sequential manner, as proposed by Wolczyk et al. (2021) for CW10. Moreover, we evaluate forgetting by measuring the performance on tasks acquired during pre-training after fine-tuning. We compare 10 different methods in this setting, as listed in Section 3.3.

We train each method for 100K steps per task in CW10, with a batch size of 256. After every 100K update steps, we switch to the next task in the sequence. After every 50K update steps, we evaluate the current model on all tasks in the task-sequence. For L2M, we use a pool size of 30. For all L2P-based approaches, we use a prompt size of 25 and a prompt pool of 100 by default. Following Wang et al. (2022c), we use the selection count $n(\mathbf{k}_i)$ up to the previous task and $\lambda = 0.5$ for regularization of L2M and all L2P-based approaches. For EWC, we tune the value of the penalty (see Section H.5). For L2M, we use learning rate of $5e^{-5}$ on CW10 and $1e^{-4}$ on DMC6. For all L2P variants, we use a learning rate of $1e^{-3}$.

We show the performance and forgetting scores for all considered methods in Table 7. All metrics are reported at the end of training. The forgetting scores are computed as defined by Wolczyk et al. (2021). In Table 7, we also include the multi-task performance scores on all CW10 tasks (same as in Appendix G). Multi-task FT represents the upper bound in terms of performance. Overall, L2M achieves the highest success rates/normalized scores and outperforms all other methods in both domains.

**Table 7:** Continual RL experiments on **(a)** CW10 and **(b)** DMC6.

(a) CW10

| Method | Success Rate | Mean Reward | Forgetting |
|---|---|---|---|
| FT | 0.09 ± 0.01 | 234.57 ± 414.86 | 0.75 ± 0.32 |
| FT-last+head | 0.12 ± 0.03 | 266.59 ± 406.42 | 0.65 ± 0.31 |
| FT-head | 0.07 ± 0.03 | 95.34 ± 127.2 | 0.08 ± 0.27 |
| L2M-oracle | 0.77 ± 0.03 | 1294.72 ± 385.32 | 0.05 ± 0.16 |
| L2M | 0.65 ± 0.04 | 1136.17 ± 53.82 | 0.07 ± 0.08 |
| L2P-Pv2 | 0.4 ± 0.02 | 707.84 ± 35.91 | 0.03 ± 0.06 |
| L2P-PreT | 0.34 ± 0.05 | 606.33 ± 61.97 | 0.01 ± 0.06 |
| L2P-PT | 0.23 ± 0.05 | 390.28 ± 64.87 | 0.11 ± 0.01 |
| EWC | 0.17 ± 0.01 | 257.48 ± 18.18 | 0.78 ± 0.0 |
| L2 | 0.1 ± 0.0 | 237.07 ± 522.98 | 0.0 ± 0.0 |
| FT-MT-scratch | 0.85 ± 0.07 | 1476.61 ± 395.13 | - |
| FT-MT-pre-trained | 0.92 ± 0.02 | 1491.76 ± 332.0 | - |

(b) DMC6

| Method | Normalized Score | Mean Reward | Forgetting |
|---|---|---|---|
| FT | 0.27 ± 0.01 | 225.81 ± 10.2 | 0.83 ± 0.02 |
| FT-last+head | 0.25 ± 0.02 | 204.49 ± 22.07 | 0.51 ± 0.06 |
| FT-head | 0.06 ± 0.0 | 58.84 ± 0.64 | -0.02 ± 0.0 |
| L2M-oracle | 0.7 ± 0.11 | 549.29 ± 254.69 | -0.02 ± 0.27 |
| L2M | 0.56 ± 0.18 | 401.65 ± 122.75 | 0.11 ± 0.14 |
| L2P-Pv2 | 0.32 ± 0.09 | 271.39 ± 72.49 | 0.02 ± 0.08 |
| L2P-PreT | 0.2 ± 0.04 | 182.33 ± 23.28 | -0.03 ± 0.01 |
| L2P-PT | 0.18 ± 0.03 | 158.09 ± 22.88 | -0.06 ± 0.02 |
| EWC | 0.3 ± 0.06 | 244.7 ± 54.12 | 0.49 ± 0.14 |
| L2 | 0.2 ± 0.03 | 192.8 ± 23.86 | -0.01 ± 0.01 |
| FT-MT-scratch | 0.92 ± 0.03 | 648.49 ± 354.7 | - |
| FT-MT-pre-trained | 1.04 ± 0.17 | 764.38 ± 230.28 | - |

## H.1 Query Representation Ablations

For all L2M results, we use embedded state-tokens aggregated over the context as query to the modulation pool. It is possible to represent the query differently by using other tokens (see Figure 8), intermediate layers, or history lengths.

**Layer Representation.** Instead of using the representation of the embedding layer for the query, we can alternatively use the representations produced by intermediate Transformer layers. To this end, we conduct an ablation study in which we compare the representations extracted from the embedding layer against the representations from the first, middle (3rd block) and last Transformer block. We show the results of this experiment in Figure 20. Using the representations extracted from the middle or last Transformer blocks yields worse performance compared to the embedding layer. In contrast, the representation from the first Transformer block attains the highest scores overall, and narrows the gap to L2M-oracle. However, it is important to note that this performance gain comes at the cost of

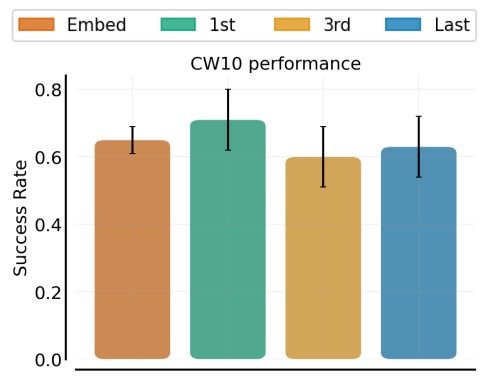

**Figure 20:** Layer representation ablation for L2M on CW10.

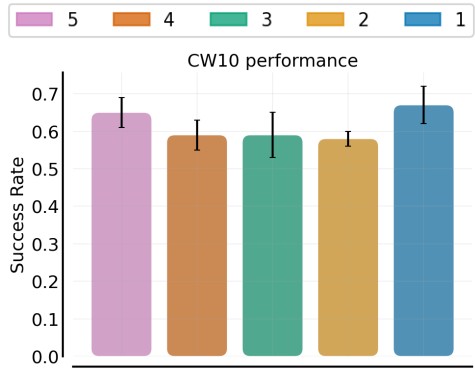

**Figure 21:** History length ablation for L2M on CW10.

additional complexity since we need to propagate farther through the model than merely through the embedding layer.

**History Length.** To investigate the dependence of the query representation on the history length while keeping the context window as is, we conduct an ablation study. In Figure 21, we report the results of this experiment. By default, we used the five last timesteps in the sequence to construct the query. However, we find that using only the most recent timestep performs similarly.

## H.2 Alternative Modulators

In L2M, we use LoRA by default. However, L2M can also be combined using other PEFT techniquesTherefore, we conduct an ablation study in which we compare L2M against L2M in combination with $(IA)^3$ on CW10. Instead of the low-rank modulation matrices, $(IA)^3$ employs element-wise multiplication with learnable modulation vectors. We present the results in Figure 22. While performance decreases with L2M-$(IA)^3$, it compares favourably in terms of parameter-efficiency. Depending on the task at hand, this may be preferred.

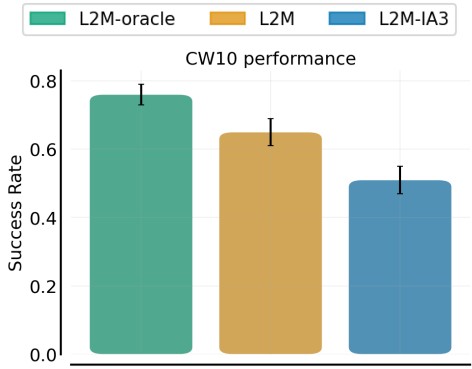

**Figure 22:** Alternative modulators ablation on CW10. We compare L2M to L2M-$(IA)^3$.

## H.3 Single-domain DT on Meta-World

We pre-train a DT with 13M parameters on MT40 and compare to the same methods as in Sections 3.2 and 3.3. In this setting, we additionally add two competitors, namely PromptDT (Xu et al., 2022b) and VIMA (Jiang et al., 2022). . We experimented with larger models, but found the selected model size to perform well. Our MDDT used a unified state space, a discretised action space and autoregressive action prediction (at inference time) to handle varying state and action spaces. In contrast, the single-domain DT does not require these mechanisms. Instead, the single-domain DT is trained to predict the continuous actions via the MSE loss. The results for the single-domain experiment are available in Figure 23. The pre-trained model attains a success rate of 81% on MT40. Overall, we find that the specialised single-domain model obtains considerably higher performance scores than the MDDT (see Sections 3.2 and 3.3). However, these performance gains come at the cost of the loss of generality, as the specialised model can only handle the particular state/action space it was trained on.

## H.4 Cross-domain FT

We conduct another experiment, in which we pre-train a DT on Meta-World only (MT40) and then fine-tune it on DMControl (DMC6). The pre-trained model relies on the unified state space and action discretisation. It has the same amount of parameters as the MDDT (40M) and attains an

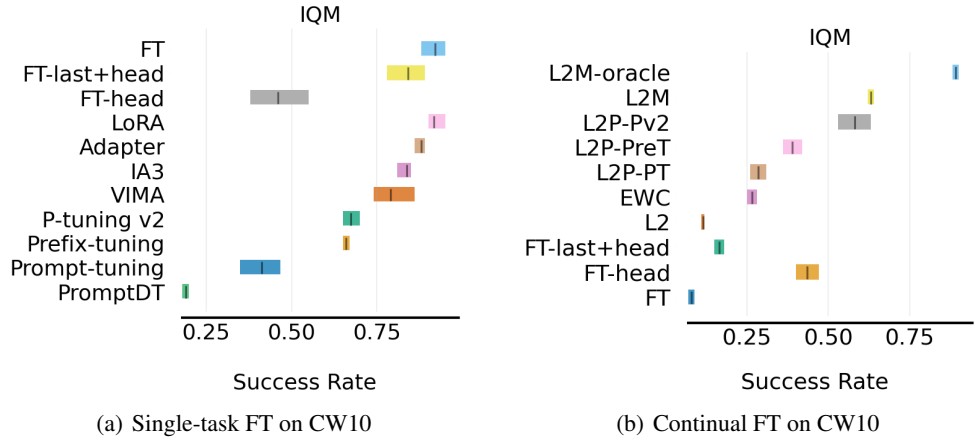

(a) Single-task FT on CW10

(b) Continual FT on CW10

**Figure 23:** IQM and 95% CIs on CW10 for **(a)** single-task FT and **(b)** continual FT using a single-domain non-discretized DT pre-trained on MT40 with 13M parameters.

average success rate of 76% on MT40 after pre-training. Prior to fine-tuning, the model has never seen trajectories from DMControl.

We show the results for different fine-tuning techniques in Figure 24 We observe that the fine-tuning performance on DMC6 is considerably worse than for the MDDT (79%). In addition, we also fine-tune the pre-trained single-domain model on the same domain (CW10). Interestingly, the final performance on CW10 is also lower compared to the MDDT model (75%) that was pre-trained on both domains. This experiment indicates, that multi-domain pre-training can, indeed, have a positive effect on the fine-tuning performance.

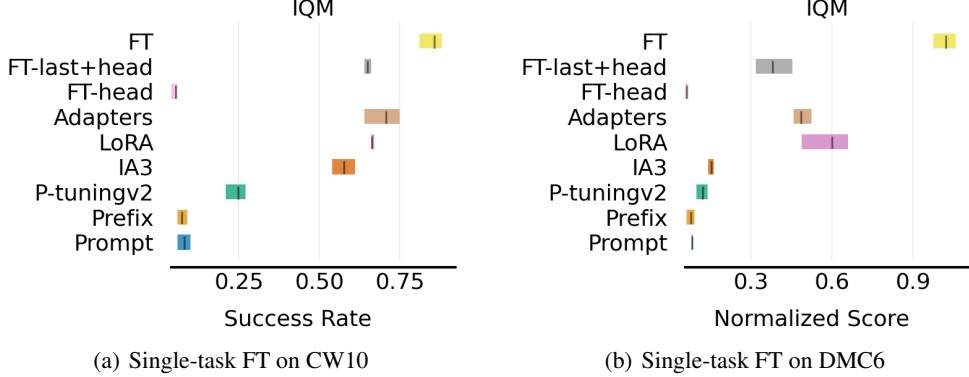

(a) Single-task FT on CW10

(b) Single-task FT on DMC6

**Figure 24:** IQM and 95% CIs for single-task FT on **(a)** CW10 and **(b)** DMC6 using a single-domain discretized DT pre-trained on MT40 with 40M parameters.

## H.5 EWC Hyper-Parameter Search

We observed that EWC (Kirkpatrick et al., 2017) performs worse than L2M in our experiments. By default, we used a regularization coefficient of $\lambda = 10000$ for EWC, which performed best in preliminary experiments. Additionally, we compare different values for $\lambda = \{10, 400, 1e^3, 1e^4, 1e^5\}$ as used in EWC. We find that the optimal choice for $\lambda$ varies heavily between the two domains, but higher performance scores can be achieved by tuning $\lambda$.

# I  Hardware & Training Times

**Pre-training**. We run all our pre-training experiments on 4 NVIDIA A100 GPUs. Training times depend on the model size. Training the smallest model (40M parameters) takes roughly 1.5 days, while training the largest model (200M) takes roughly 3.5 days. To parallelize the computation across multiple GPUs/nodes, we leverage the distributed data parallel (DDP) feature provided by PyTorch. Throughout all our experiments, we use mixed precision training (Micikevicius et al., 2017) as supported in PyTorch to speed up training time.

**Fine-tuning**. For all our fine-tuning experiments, we use single GPU training on NVIDIA A100 or NVIDIA Titan V GPUs. Training times vary between the considered fine-tuning settings (single-tasks experiments, continual-learning experiments), domains (Meta-World, DMControl), and methods. For example, running a single seed for L2M on CW10 in the continual FT setting takes 1.25 days on one NVIDIA A100. Similarly, running a single seed for L2M on DMC6 takes roughly one day on one NVIDIA Titan V.

# J  Potential Societal Impact

Our work is purely academic. However, the presented ideas aim for more capable embodied agents. As such, there is potential for misuse, e.g., when used in real-world scenarios without proper verification or safeguards. While we do not expect to see the deployment of such agents for potentially malicious purposes in the foreseeable future, it is essential to ensure responsible use of this technology.

