# OpenReview forum: "Learning to Modulate pre-trained Models in RL"
_NeurIPS.cc/2023/Conference — NeurIPS 2023 poster_

### Official Review · Reviewer_aphp · 2023-06-28

**Soundness:** 3 good
**Presentation:** 3 good
**Contribution:** 3 good
**Rating:** 7
**Confidence:** 3

**Summary:**

Large-scale pretraining on a diverse dataset followed by finetuning on smaller datasets from downstream tasks has been wildly successful in domains such as computer vision and NLP. The closest analogue to this paradigm in the context of RL is arguably multi-task pretraining followed by finetuning on one or more unseen tasks. This paper investigates the efficacy of finetuning approaches popularized by supervised learning (CV/NLP) on RL problems cast as sequence modeling with Decision Transformers (DT). The authors construct a pretraining dataset that consists of 50 state-based tasks from Meta-World and DMControl (40 and 10 tasks, respectively), and evaluate finetuning methods on held-out tasks from each domain (10 and 6 tasks, respectively) in both single-task finetuning, multi-task finetuning, and continual learning settings. The authors find that their proposed finetuning method, L2M, which combines L2P and LoRA, consistently obtains good performance on unseen tasks after finetuning, while also retaining good performance on the pretraining tasks.

**Strengths:**

The problem is interesting, paper is well written and easy to follow, the method is well motivated, and experiments appear sound. Sufficient discussion of related work. While many existing papers have considered multi-task pretraining and finetuning in RL, I appreciate that the authors take the time to thoroughly investigate trade-offs between different finetuning methods. Further, new finetuning strategies such as LoRA have become very popular in NLP, and this paper confirms that it (along with other modifications necessary to make it work for multi-task DTs as proposed by the authors) can also work well for DTs.

**Weaknesses:**

- **Lack of clarity on experimental setup.** When going through the paper, I found it difficult to fully grasp what the experimental setup looks like and its potential assumptions / pitfalls / failure modes without repeatedly checking the appendix and/or reading between the lines. For example, it is not stated explicitly that finetuning is done strictly on offline replay data which is also collected by single-task SAC agents as in the pretraining dataset. I had to find this information in Appendix D. Likewise, it is not stated explicitly which tasks are included in the pretraining and finetuning datasets, I had to find this information in Appendix A. For the former, it is not really a problem that finetuning requires an offline dataset for the target task, but not making it clear is deceiving. For the latter, the authors do mention that the DMControl tasks include multiple embodiments but do not provide further details in the main paper. I find this problematic since different splits and/or sets of tasks would lead to very different finetuning performances (e.g. task difficulty and degree of overlap).

- **Lack of discussion on limitations.** Continuing along the lines of the above, I also find that the paper generally lacks discussion of limitations. Given that the work is very data-driven and domain gaps generally are larger in RL than in NLP, it is important to clearly state assumptions / pitfalls / failure modes related to data collection and experimental setup. Ideally, these limitations (or properties, if you will) would be backed by data that shows, e.g., that finetuning is highly dependent on task similarity. The authors list this as future work but adding such an experiment would make the current submission more complete. For example, the authors could pretrain on Meta-World and finetuning on DMControl, and vice versa, and compare to the performance when including same-domain tasks in the pretraining dataset.

**Questions:**

I would like the authors to address the comments I listed in "weaknesses". Additionally, two clarification questions:

- It is stated in Appendix A that the task *reacher-hard* is included in both the pretraining and finetuning datasets. I assume that this is written in error, but would like the authors to please list the correct task splits.

- The authors also state in Appendix A that the action spaces considered in DMControl range from 1 (cartpole) to 21 (humanoid) dimensions. However, I do not see any experiments on humanoid in the paper. Can the authors please clarify if they consider humanoid tasks or not?



**Limitations:**

I would like to see more discussion on limitations. See my previous comments for constructive feedback.

---

> ### Author Rebuttal · Authors · 2023-08-09
>
> Thank you for your positive assessment of our paper and your feedback!
>
> **Lack of clarity on the experimental setup:** Following your feedback, we revised our paper to improve clarity. In particular, we changed the following:
>
> * In Line 181 (Experiments), we now explicitly point out that the fine-tuning is done strictly on offline replay data, similar to our pre-training setup. We agree that this information should have been more prominent in the main text, rather than being mentioned in the Appendix only.
> * We added a Table to Appendix A, in which we explicitly list the state and action spaces for all pre-training and fine-tuning tasks considered in this work. While the Meta-World benchmark contains a single robot morphology (same state/action spaces across tasks), the morphologies in DMControl vary across tasks (different state/action spaces). In addition, we refer to this Table in Section 2 and give illustrative examples for Cheetah and Walker in the main text.
> * In case of acceptance, we will use the additional page, to include further details in the main text, that are currently relegated to the Appendix due to space constraints.
>
> **Lack of discussion on limitations:**
>
> * We agree with the reviewer, that pre-training on one domain, and fine-tuning on another is an interesting experiment. Therefore, we pre-trained a DT (with action discretization, unified state space and autoregressive action prediction) on Meta-World (MT40) only and then fine-tuned it to DMC6 and CW10. We find that the single-domain model (MT40) performs worse than our multi-domain model (MT40+DMC10), both on CW10 and on DMC6 (see Figure 3 in the attached pdf). These results indicate that multi-domain pre-training has a positive effect on the fine-tuning performance. We will add these two experiments, including a more detailed discussion of the setup/results/limitations, to our manuscript.
> * Another limitation of multi-domain pre-training in RL, is that it requires discretization and autoregressive action prediction to handle varying action spaces. Therefore, we pre-trained a non-discretized DT (trained via MSE loss) on MT40 only and then fine-tune on CW10. This results in considerably higher performance scores, but comes at the cost of the loss of generality. This single-domain model can only handle the particular state/action space it was trained on, and thus, fine-tuning it to tasks with new state/action spaces is not possible. We added a discussion on these limitations in our paper.
>
> Regarding your **questions**:
>
> * Thank you for spotting this, and sorry for the confusion. You are right, this was an error, and we already corrected it. Reacher-hard is in the pre-training dataset, and reacher-easy is in the fine-tuning dataset. We selected our 6 fine-tuning tasks for DMControl in line with prior work (Hafner et al., 2019; Yarats et al., 2020).
> * As stated in lines 705-706, the action spaces in DMControl range from 1 (cartpole) to 21 (humanoid) across all tasks. However, in the 16 DMControl tasks we select (see Appendix A) the action spaces vary between 1 (cartpole, pendulum) and 6 (cheetah, walker). Humanoid is not among these 16 tasks. For clarification, we added a table in Appendix A that lists the action spaces (and original state spaces) for all environments we consider in our experiments.
>
> Thank you again for your actionable suggestions. We added the clarifications and experiments, including a discussion of respective limitations, to our manuscript. If there are any further questions, we would be happy to discuss them!
>
> **References**:
>
> * Hafner, Danijar, et al. "Learning latent dynamics for planning from pixels." International conference on machine learning. PMLR, 2019.
> * Yarats, Denis, Ilya Kostrikov, and Rob Fergus. "Image augmentation is all you need: Regularizing deep reinforcement learning from pixels." International conference on learning representations. 2020.

---

> > ### Comment · Reviewer_aphp · 2023-08-17
> > **Thank you**
> >
> > Thank you for the clarifications and additional experiments. I believe that this paper will be useful to the NeurIPS community, and incorporating all of the feedback you have received during this rebuttal (from fellow reviewers and I) into a future revision will further strengthen it. I have raised my score accordingly.

---

### Official Review · Reviewer_yPGU · 2023-07-04

**Soundness:** 3 good
**Presentation:** 2 fair
**Contribution:** 4 excellent
**Rating:** 5
**Confidence:** 4

**Summary:**

This paper studies fine-tuning and continual learning of pre-trained decision transformers in RL. Extensive experiments are conducted to analyze naive fine-tuning, parameter-efficient fine-tuning, and prompt tuning methods on both Meta-world and DMC domains. This paper presents a new method L2M, which combines well-established LoRA and L2P methods and demonstrates the superiority of L2M on Continual-World and DMC benchmarks.

**Strengths:**

1. The proposed method is well-motivated and carefully designed to enable a general agent on multiple domains and tasks.
2. Extensive experiments and ablation study on modulating pre-trained DT
3. Strong performance on the continual learning benchmark

**Weaknesses:**

1. The most significant weakness is the relatively poor presentation of the methodology and experiments, mainly due to the absence of many details. See questions below.
2. Although a unified state space is manually designed for MDDT in this paper, if I understand correctly, it is hard to extend it for new domains with a distinct state space, such as RLBench.

**Questions:**

On the methodology:

1. According to Eq. (2) and (3), it seems that each step of DT separately determines a distinct choice of the modulator. Does this mean that we have different weights in Eq. (1) for each token in the sequence? If so, can it break the advantage of training in parallel for transformers?
2. In Eq. (3), how do we update n(k)? What do we mean by selection count? Do we add n(k) by one, once we encounter a trajectory selecting k?
3. In Line 132, how the learnable keys are updated? I cannot find this additional term in the main text or appendix.
4. In Line 359, the authors state that future work includes combining and sharing modulators across tasks. However, if I understand correctly, we have already shared the modulation pool across tasks in the continual learning setting.

On the experiments:

5. In Line 208, the authors claim that they also experimented with PrompDT and VIMA, but I cannot find any experimental details or results.
6. In Section 3.2, where are the details about L2M-oracle? What kind of information (e.g., textual task specification?) is provided to L2M, and how is this information provided in the model? I cannot find implementation details either.
7. In Figure 6, it seems there is a line of straightforward baselines, i.e., separately training a new LoRA or adapter for each new task. Why are these baselines not evaluated?
8. In Figure 7, why do PEFT methods hurt performance on the pretrained tasks? If I understand correctly, we freeze all the parameters of the pre-trained model and only fine-tune the modulators.

Minor suggestions on presentation:

9. Since L2M combines L2P and LoRA, the authors should present L2P in the Background section, provide intuitions on how L2P mitigates forgetting, and highlight the difference between L2P and L2M (one uses prompt tuning and one uses LoRA).
10. In Figure 4, the captions of subfigures should be CW10 and DMC6, respectively.

Overall, I appreciate the effort made by the authors to conduct extensive experiments and carefully design the method. I recommend the author continuously improve this paper and make it impactful.

If the authors solve my questions properly and plan to revise their presentation, I will be happy to increase my rating.

------

Update: The authors have responded with detailed clarification in their rebuttal and most of my concerns are addressed. Thus I increase my rating from 4 to 5.

**Limitations:**

This work has discussed its limitations, future work, and social impacts. I recommend including weakness 2 mentioned above in the limitation part.

---

> ### Author Rebuttal · Authors · 2023-08-09
>
>
> Thank you for your excellent feedback, it helped us to considerably improve our paper! We conducted additional experiments (see attached PDF). In the revised manuscript, we incorporated all your feedback and suggestions.
>
> **Presentation of methodology:**
> 1. **Training in parallel:** At training time, the modulation matrices are selected for the entire sequence, not on a per-step basis. Thus, our method does not break the advantage of training Transformers in parallel. Thanks for pointing this out, we will highlight this in the methods section of the paper.
> 2. **Selection count n(k):** The selection count n(k) refers to the keys that map to the modulators in the modulation pool. This means that during training time, we maintain a count of how often a given key (and respective modulator) has been selected up until the current task. Once a key is selected, we increase its count n(k) by 1. To discourage L2M from always selecting the same modulators, we use the inverse of the selection count in Equation 3.
> 3. **Learnable keys:** For updating the learnable keys, we employ the same update strategy as L2P. This means, we use a surrogate loss term to pull the selected keys closer to the corresponding query features by maximizing their cosine similarity. This loss term is added to the regular cross-entropy objective for action prediction used by the Decision Transformer. We now include the exact equation in the paper.
> 4. **Combining modulators:** You are correct, we are sharing the modulation pool across tasks in the continual learning setting. However, for a given input query, only a single set of modulation matrices is selected. What we refer to in Line 359 is that it may be possible to select multiple suitable modulators for a given input query and compose them accordingly. We understand that this distinction was not formulated clearly enough, and revised our formulation accordingly in our manuscript. For example, we refined our wording in Line 361 by omitting the phrase “sharing modulators”, because as you rightly pointed out, the modulation pool is already shared across tasks.
>
> **Presentation of experiments**:
>
> 5. **PromptDT and VIMA:** Yes, we did conduct experiments with PromptDT (Xu et al., 2022) and VIMA (Jiang et al., 2022). We experimented with them in a single-task setting, in which we trained on Meta-World only. Therefore, we now included the results for the single-task setting in our final version (see Figure 2 in the attached PDF).
> 6. **L2M-oracle:** Thanks for highlighting this. L2M-oracle obtains the information on what task is currently being observed. We state in line 252: “Moreover, we add another implementation of L2M, which is equipped with an oracle that provides information on what task is currently being observed”. Following line 252, we added a more concrete explanation of L2M-oracle in our methods section:
>     * “This information is provided in terms of the task index. For L2M-oracle, the modulation pool contains as many modulators as there are tasks. The given task index is then used to select the respective modulators. At training time, the task index refers to the dataset the batches are sampled from. At inference time, the task index refers to the environment the DT is currently evaluated in.”
> 7. **Baseline:** In fact, the suggested baseline is exactly the L2M-oracle baseline, which trains a single set of LoRA weights per task (see previous point).
> 8. **PEFT on pre-training tasks:** You are right, the pre-trained model is frozen and only the modulators are fine-tuned, thus the performance on pre-training tasks is not affected by PEFT. However, to remain task-agnostic, we train a set of 100 additional keys (see answer to question 3) on the pre-training datasets. At inference time, we concatenate this set to the set of keys introduced by L2M during continual fine-tuning. Therefore, the model does not need to be told if the inputs come from a pre-training or fine-tuning task. If a “pre-training key” is selected, no modulation occurs. The slight performance drop in Figure 7 comes from conflation effects between the pre-training and fine-tuning keys. We updated our manuscript with this additional information.
>
>
> **Minor suggestions:**
>
> 9. **Background on L2P:** Thank you for this suggestion. We now include a more detailed description of L2P and distinction to L2M in the background section. Among others, we added additional information on:
>     * The selection of the modulation matrices (per step vs. per sequence).
>     * The usage of the selection count penalty.
>     * The exact loss function to optimize the key.
> 10. Thanks for pointing this out, we have fixed this mistake.
>
> **Limitations of state-space:** The designed state-space is indeed specific to the benchmarks considered in our work. We are aware of this limitation and will add a discussion on this point to our paper. We believe that other benchmarks with differing state spaces, such as RLBench, can be approached similarly. However, we are planning to explore alternative approaches in future work.
>
> Overall, we are very grateful for your extensive suggestions. We revised our paper, and we believe that your suggestions improve our paper. Furthermore, we added additional experiments (as included in the one-page PDF), to provide further empirical support.
>
> **References**:
> * Mengdi Xu, Yikang Shen, Shun Zhang, Yuchen Lu, Ding Zhao, Joshua Tenenbaum, and Chuang Gan. Prompting decision transformer for few-shot policy generalization. In International Conference on Machine Learning, pp. 24631–24645. PMLR, 2022.
> * Yunfan Jiang, Agrim Gupta, Zichen Zhang, Guanzhi Wang, Yongqiang Dou, Yanjun Chen, Li Fei-Fei, Anima Anandkumar, Yuke Zhu, and Linxi Fan. Vima: General robot manipulation with multimodal prompts. arXiv preprint arXiv:2210.03094, 2022.

---

> > ### Comment · Reviewer_yPGU · 2023-08-11
> >
> > I would like to express my appreciation for the detailed response, which has addressed most of my concerns.
> >
> > I have also read other reviews and responses and found Reviewer Aphp also concerns about the clarity. Given that the authors have responded with helpful clarification and made a revision (though I cannot see it due to the policy of NeurIPS this year), I decided to increase my rating to 5.

---

### Official Review · Reviewer_Mf3J · 2023-07-04

**Soundness:** 3 good
**Presentation:** 2 fair
**Contribution:** 3 good
**Rating:** 7
**Confidence:** 3

**Summary:**

The authors study the problem of preventing catastrophic forgetting in DT finetuning. The proposed method leveraged a pool of LORA adaptors and only choose the relevant adaptor matrix during finetuning. The author achieve good results on continual world.

**Strengths:**

The applicaiton of lora pools for finetuning DT is novel and the problem of continual learnning in DT is important. The experiment results show that the proposed method works and the released dataset should have a good impact to the community.

**Weaknesses:**

The only weakness I think is the presentation of the method. It seems that the paper is largely inspired by Learing to prompt, and as a result, I believe many of the technical details are not explained in here. E.g., the exact loss function to optimize key. Section I should be polished to include more explanation of the method.

**Questions:**

N/A

---

> ### Author Rebuttal · Authors · 2023-08-09
>
> Thank you for your feedback, which helped a lot to improve our manuscript. We appreciate your positive assessment of our work: thank you. We are optimistic that our dataset will contribute to advance the RL research community.
>
> Thank you for pointing out the lack of technical details regarding our method. In the revised manuscript, we now included more details and elaborated much more on them:
> * The exact loss function to optimize the key.
> * The selection of the modulation matrices (per step vs. per sequence).
> * The usage of the selection count penalty.
>
> To bring more details indeed improved our manuscript a lot.

---

### Official Review · Reviewer_PZAn · 2023-07-06

**Soundness:** 3 good
**Presentation:** 3 good
**Contribution:** 2 fair
**Rating:** 5
**Confidence:** 3

**Summary:**

The authors propose an adaptation method for pretrained DT (decision transformer) that combines two finetuning techniques, learning-to-prompt and low rank adaptation (L2P + LoRA), which have been investigated in NLP and computer vision domains. This combined method aims at exploiting the benefits of finetuning and prompt-based learning so that adaptation can be achieved parameter-efficiently and without much catastrophic forgetting.

**Strengths:**

The authors provide the evaluation and comparison on finetuning technique applications to DT-based RL policies, including full finetuning, finetuning with action head, adapters, LoRA, and prompt-tuning, and prefix-tuning that have been well investigated in NLP and computer vision domains.

Based on the evaluation, the authors propose an adaptation method combining L2P and LoRA, by which the pretrained DT can be used to solve new tasks.

**Weaknesses:**

The evaluation of different finetuning techniques as survey is meaningful and helpful for readers, but the proposed solution simply combines the two techniques, and little analysis has been conducted on it. There might be some other combination based on L2P, e.g., L2P with IA3, for adaptation.

The authors do not explain clearly why multi-domain DT (MDDT) is considered, e.g., MDDT can be useful and effective for learning each domain through shared knowledge and representation.
Does the proposed L2P+LoRA get benefited from MDDT? What if a single domain DT was tested?

Minor errors:
- In line 21, no Section 1 title.
- In Figure 4(a), the caption should be CW10.
- In Figure 4(b), DMC10 should be DMC6.
- In Figure 6(b), Success rate should be Normalized score.
- In line 326, a missing citation.
- Some citation forms are incorrect.

**Questions:**

Are there any other finetuning techniques that can be combined with L2P, similar to L2P with LoRA?

HyperDT [1] handles parameter efficiency, where LoRA part is similar. Could the authors compare the proposed solution with HyperDT?

[1] Xu, Mengdi, et al. "Hyper-decision transformer for efficient online policy adaptation." arXiv preprint arXiv:2304.08487 (2023).

**Limitations:**

No specific statement on limitations.

---

> ### Author Rebuttal · Authors · 2023-08-09
>
> Thank you for your helpful feedback. Our manuscript improved considerably by addressing and incorporating your comments.
>
> **Analysis:** We are glad that you find the evaluation of fine-tuning techniques meaningful and helpful for readers. Regarding additional analysis on L2M, we already investigated:
> * the effect of the rank for LoRA (see Figure 17 in Appendix G).
> * the choice of embedding tokens for the query in L2M (Figure 19 in Appendix F). As per suggestion of another reviewer, we further expanded this ablation study (Figure 1a in the attached PDF).
> * the selection of modulation targets in L2M (Figure 20 in Appendix F).
> * as per suggestion of another reviewer, we now also added an investigation in which we vary the embedding history length used to construct the query in L2M (Figure 1b in the attached PDF).
>
> We agree that further combinations such as L2M with IA3 (Liu et al., 2022) are of interest for adaptation. In Figure 18, Appendix F, we report the results of this comparison. While using IA3 performs worse, it compares favorably in terms of parameter efficiency. In addition, we provided results for L2P in combination with other prompt-tuning based approaches, such as L2P+Pv2 and L2P+PreT (see Figure 18 in Appendix F). In the revised manuscript, we now highlight these variations more prominently.
>
> **Multi-domain DT:** Thank you for pointing this out to motivate the multi-domain setting and why it is highly relevant in RL. In the revised manuscript, we now motivate our multi-domain setting in much more detail. We want to emphasize that L2M is independent of the pre-training paradigm and applicable in both single and multi-domain scenarios. Overall, we believe that the more challenging multi-domain setting is also more interesting for practitioners as it is a more realistic scenario. Having said that, we do agree that a more thorough investigation of the multi-domain setting does strengthen our paper considerably. Therefore, we performed the following additional experiments:
> * **Single-domain (Figure 2 in the attached PDF):** pre-training and fine-tuning only on Meta-World (MT40+CW10). Due to the common state and action spaces, we used a simpler (non-discretized) action space and training objective (MSE) for this experiment. This results in considerably higher performance scores. However, this comes at the cost of the loss of generality. The single-domain model can only handle the particular state/action space it was trained on. Thus, fine-tuning it to tasks with new state/action spaces is not possible.
> * **Cross-domain fine-tuning (Figure 3 in the attached PDF):** pre-training on Meta-World only (MT40) and fine-tuning on DMControl (DMC6). Here, we use the same discretized action space as for MDDT to account for different action spaces across domains. Overall, we observe that the fine-tuning performance on DMC6 (different domain) is considerably worse than for the MDDT (Figure 3b). In addition, we also fine-tune the pre-trained single-domain model on CW10 (same domain). The final performance on CW10 is also lower compared to the MDDT model that was pre-trained on both domains. These experiments indicate, that multi-domain pre-training, indeed, has a positive effect on the fine-tuning performance (Figure 3a).
>
> **Other fine-tuning techniques:** As discussed above, we agree that investigating other fine-tuning techniques in combination with L2M is interesting, and already conducted this investigation in our submission. Due to space constraints, these comparisons have been relegated to the appendix, see Figure 18 in Appendix F.
>
> **HyperDT:** Thank you for suggesting the comparison against HyperDT (Xu et al., 2023). Unfortunately, there is no open-source implementation available yet, but we are working on a reimplementation to include HyperDT in our comparison. We expect HyperDT to work well in the single-task setting, but to fall behind in the continual setting, as it has no mechanism to prevent forgetting.
>
> We hope to have clarified all your questions. We revised our paper to highlight the conducted ablation studies, included our additional experiments in the single-domain setting, and aim to add the HyperDT baseline. In case any questions remain, we would be happy to answer them!
>
> **References**:
> * Liu, Haokun, et al. "Few-shot parameter-efficient fine-tuning is better and cheaper than in-context learning." Advances in Neural Information Processing Systems 35 (2022): 1950-1965.
> * Xu, Mengdi, et al. "Hyper-decision transformer for efficient online policy adaptation." arXiv preprint arXiv:2304.08487 (2023).

---

> > ### Comment · Reviewer_PZAn · 2023-08-20
> >
> > I'd like to extend my thanks for the comprehensive response, which addresses the most of the concerns I had, particularly regarding the motivation behind multi-domain DT in this work and the discussion on other fine-tuning techniques with IA3 in the appendix. I would be inclined to raise my score slightly, given these explanations.

---

### Official Review · Reviewer_tSZN · 2023-07-07

**Soundness:** 3 good
**Presentation:** 2 fair
**Contribution:** 2 fair
**Rating:** 6
**Confidence:** 3

**Summary:**

The paper considers the catastrophic forgetting problem in pre-training and fine-tuning RL setting.  The paper proposes Learning-to-Modulate (L2M) to reduce the degradation of pretrained models by modulating the information flow of the frozen pre-trained model via a learnable modulation pool.

L2M shows state-of-the-art performance on a continual learning benchmark, while retaining performance on the pre-training tasks.

**Strengths:**

+ The paper is overall well written and easy to follow.
+ The paper proposes L2M which is a parameter-efficient fine-tuning and prompt-based tuning prompting method. The method is sound and achieves good results.

**Weaknesses:**

- Enhanced baselines: How does L2M measure up against the latest advancements in training methods, such as https://arxiv.org/abs/2211.12740, https://arxiv.org/abs/2301.09816, https://arxiv.org/abs/2211.10869, and https://arxiv.org/abs/2305.16554? These techniques have demonstrated enhanced training of Transformers, resulting in improved generalization and scalability across numerous tasks. Considering the success of these methods in scaling up pretraining, does L2M still hold its relevance?
- Absence of weight decay baseline: In the paper, the authors discuss the limitations of commonly used methods that encounter catastrophic forgetting during continual learning. However, it is worth investigating whether the authors explored the effectiveness of weight decay as a preventive measure against overfitting to new tasks. Weight decay, being a straightforward yet potent technique, has proven effective in mitigating overfitting during finetuning processes.

Update: The authors have responded with detailed clarification in their rebuttal and most of my concerns are addressed. Thus I increase my rating to 6.

**Questions:**

Is the embedding history state only, did the author consider state-action history?

How does the embedding history size impact the results?

**Limitations:**

Yes, the authors adequately addressed the limitations.

---

> ### Author Rebuttal · Authors · 2023-08-09
>
> Thank you for your constructive feedback and suggestions that improved our manuscript.
>
> **Enhanced baselines:** We agree that the methods, to which the reviewer is referring, are relevant for improving the pre-training stage. However, our method aims at improving the fine-tuning phase, where it prevents forgetting of already learned tasks. Therefore, the referred methods seem orthogonal and might be used in combination with our method to improve overall performance. Consequently, we leave exploring this combination of methods for future work.
>
> **Weight decay:** Indeed, we explored weight decay via L2 regularization on weights of previous tasks (see Figure 6 in the manuscript). Weight decay has been investigated by the community for preventing catastrophic forgetting. However, weight decay was found to be inferior to EWC, which we included as a baseline in Figure 6 (Kirkpatrick et al., 2017). Furthermore, weight decay was explored in relation to loss of plasticity and found to be ineffective in its naive form (Dohare et al., 2021; Lyle et al., 2023). We added a discussion regarding weight decay to the revised manuscript, where we elaborate more on the L2 weight decay baseline.
>
> **History embedding:** In our experiments, the embedding history only contained state tokens. However, we conducted an ablation study on how to represent the history in Figure 19 in Appendix H. In Appendix H we compare various options and combinations of state/action/reward/RTG tokens to represent the history. We agree with the reviewer that a combination of state and action embeddings must be investigated. Therefore, in the revised manuscript, we now provide results for state-action history as well as state-RTG history (see Figure 1a in the attached PDF). As expected, the state token embeddings perform best.
>
> **History size:** The reviewer is right, the dependence on the history size is an important question. In the revised manuscript, we now include an ablation study, where we vary the embedding history size (see Figure 1b in the attached PDF). By default, we used the embedded state tokens of the 5 last timesteps to construct the query.
>
> Our manuscript profited a lot by incorporating your suggestions: thank you.
>
> **References:**
> * Kirkpatrick, J., Pascanu, R., Rabinowitz, N., Veness, J., Desjardins, G., Rusu, A. A., Milan, K., Quan, J., Ramalho, T., Grabska-Barwinska, A., et al. (2017). Overcoming catastrophic forgetting in neural networks. Proceedings of the national academy of sciences, 114(13):3521–3526.
> * Dohare, Shibhansh, Richard S. Sutton, and A. Rupam Mahmood. "Continual backprop: Stochastic gradient descent with persistent randomness." arXiv preprint arXiv:2108.06325 (2021).
> * Lyle, Clare, et al. "Understanding plasticity in neural networks." arXiv preprint arXiv:2303.01486 (2023).

---

> > ### Comment · Reviewer_tSZN · 2023-08-11
> > **Rebuttal Acknowledged**
> >
> > I would like to thank the authors for their effort during the rebuttal.
> >
> > I appreciate the clarification on weight decay baselines and running extra experiments to ablate history size.

---

### Author Rebuttal · Authors · 2023-08-09

Dear Reviewers,

We thank you for your helpful comments, excellent feedback, and generally positive responses! We carefully read your constructive reviews and responded to all your questions and comments. Furthermore, we conducted additional experiments and report the results in the attached PDF. The revised manuscript includes much more details.

Thanks again and best regards,\
The Authors

---

### Decision · Program_Chairs · 2023-09-21

**Decision:**

Accept (poster)

**Comment:**

Within the domain of reinforcement learning, this paper tackles the challenging issue of catastrophic forgetting during fine-tuning of pre-trained models on new tasks. The reviewers unanimously acknowledge the strength of the proposed Learning-to-Modulate (L2M) method, which introduces a novel approach of modulating the information flow using a learnable modulation pool. This approach is praised for its effectiveness and innovation, particularly its application as a parameter-efficient fine-tuning and prompt-based tuning method.

The reviewers emphasise the paper's significance in addressing the ongoing problem of continual learning in deep reinforcement learning (DRL). The results from the experiments are convincing, demonstrating the success of the L2M method in achieving state-of-the-art performance on the Continual-World benchmark while preserving performance on pre-training tasks. The release of a comprehensive dataset covering tasks from Meta-World and DMControl is seen as a valuable contribution.

For the camera ready, authors should ensure they address reviewers suggestions relating to clarify experimental setup, discuss additional limitations, and address minor formatting issues.